# *Mycobacterium tuberculosis* induces decelerated bioenergetic metabolism in human macrophages

Bridgette M Cumming[1], Kelvin W Addicott[1], John H Adamson[1], Adrie JC Steyn[1,2]*

[1]Africa Health Research Institute, Durban, South Africa; [2]Department of Microbiology, Centers for AIDS Research and Free Radical Biology, University of Alabama at Birmingham, Birmingham, United States

**Abstract** How *Mycobacterium tuberculosis* (*Mtb*) rewires macrophage energy metabolism to facilitate survival is poorly characterized. Here, we used extracellular flux analysis to simultaneously measure the rates of glycolysis and respiration in real time. *Mtb* infection induced a quiescent energy phenotype in human monocyte-derived macrophages and decelerated flux through glycolysis and the TCA cycle. In contrast, infection with the vaccine strain, *M. bovis* BCG, or dead *Mtb* induced glycolytic phenotypes with greater flux. Furthermore, *Mtb* reduced the mitochondrial dependency on glucose and increased the mitochondrial dependency on fatty acids, shifting this dependency from endogenous fatty acids in uninfected cells to exogenous fatty acids in infected macrophages. We demonstrate how quantifiable bioenergetic parameters of the host can be used to accurately measure and track disease, which will enable rapid quantifiable assessment of drug and vaccine efficacy. Our findings uncover new paradigms for understanding the bioenergetic basis of host metabolic reprogramming by *Mtb*.

DOI: https://doi.org/10.7554/eLife.39169.001

*For correspondence:
adrie.steyn@ahri.org

Competing interests: The authors declare that no competing interests exist.

## Introduction

Mechanisms underlying the pathogenesis induced by *Mycobacterium tuberculosis* (*Mtb*), the etiological agent of tuberculosis (TB), are poorly understood, and increasing evidence suggests that *Mtb* subverts the host's immune response to establish a persistent infection (*Cambier et al., 2014*; *Hmama et al., 2015*; *Józefowski et al., 2008*). Crucial to the success of the immune system to control microbial infection is the metabolic plasticity of immune cells to activate antimicrobial mechanisms in macrophages and activate T cells in response to microbial invasion. Precise coordination between diverse metabolic pathways underlies this plasticity (*Ganeshan and Chawla, 2014*; *Loftus and Finlay, 2016*; *Mathis and Shoelson, 2011*), which is disrupted by pathogenic bacteria. Hence, host-directed therapies are increasingly considered for adjunctive treatment of tuberculosis (*Guler and Brombacher, 2015*; *Mahon and Hafner, 2015*; *Wallis and Hafner, 2015*).

Studies suggest that *Mtb* pathogenicity is reinforced with participation of metabolic pathways from the host, including evidence suggesting that *Mtb* adaptation to the host environment requires catabolism of host-derived lipids (*Daniel et al., 2011*; *Muñoz-Elías and McKinney, 2005*; *Pandey and Sassetti, 2008*; *Rohde et al., 2012*; *Lee et al., 2013*). This is assumed to be induced through *Mtb* regulating metabolic thresholds of the host macrophage (*Mehrotra et al., 2014*). Recent studies suggested that there is a shift from oxidative phosphorylation towards glycolysis in macrophages infected with an avirulent strain (H37Ra) or dead γ-irradiated *Mtb* (*Gleeson et al., 2016*), and in *Mtb* (H37Rv)-infected mouse lungs using transcriptomic profiling and confocal imaging (*Shi et al., 2015*). Lachmandas et al. (*Lachmandas et al., 2016*) demonstrated that the switch to

aerobic glycolysis observed in human peripheral blood mononuclear cells stimulated with dead *Mtb* lysate is TLR2-dependent, and is mediated in part through the AKT-mTOR (mammalian target of rapamycin) pathway. While this evidence supports the conclusion that dead *Mtb* reprograms host energy metabolism, the actual underlying mechanisms with live virulent *Mtb* infection enabling it to persist in humans remain elusive. Furthermore, the metabolic health of the *Mtb*-infected cell is poorly defined as there is a lack of knowledge on exactly what metabolic health comprises, and what should be measured. Thus, development of a technological advance to address these gaps in our knowledge is expected to uncover the fundamental role of host energy metabolism in allowing *Mtb* to persist for decades without causing disease.

Aberrant cellular bioenergetics have been associated with, and are often the cause of, diseases such as diabetes, cancer, neurodegeneration, and cardiac disease. The dysfunctional energy metabolism in these diseases has been successfully investigated using extracellular flux (XF) analysis (*Devarajan et al., 2011*; *Hill et al., 2009*; *Salabei et al., 2016*; *Wu et al., 2007*; *Lee et al., 2017*; *Cronin-Furman et al., 2013*). XF analysis monitors the rate of oxygen consumed by cells (oxygen consumption rate, OCR) and the release of protons from the cells into the extracellular medium (extracellular acidification rate, ECAR) non-invasively in real time (*Figure 1A*). Measurements of cellular respiration and acidification form the foundation of our understanding of bioenergetics because cells use two main pathways to produce ATP, namely oxidative phosphorylation (OXPHOS) and glycolysis. This technology is largely unexplored in the field of bacterial pathogenesis, with a few studies focused on *Helicobacter pylori* infections (*Hammond et al., 2015*; *Saha et al., 2010*), but studies on live virulent *Mtb* pathogenesis are lacking.

In this study, we used extracellular flux analysis to explore the modulation of the energy metabolism of differentiated THP-1 macrophages and human monocyte derived macrophages (hMDM) infected with live virulent *Mtb*, the slow-growing non-pathogenic vaccine strain, *M. bovis* BCG (BCG) and dead-*Mtb.* We examined how mycobacterial burden affects OXPHOS and the glycolysis of macrophages, we investigated ATP production by glycolysis and OXPHOS during mycobacterial infection, and assessed the capacity, dependency and flexibility of mitochondria on glucose, glutamine or fatty acids during infection. Lastly, we confirmed our findings with [U-$^{13}$C]glucose stable isotope tracing experiments. By adapting a real-time, non-invasive bioenergetic platform to study the bioenergetics of the *Mtb*-infected host cell, we have generated new knowledge that may contribute towards a better understanding of *Mtb* persistence and development of novel approaches for host-directed therapeutic interventions.

## Results

### *Mtb* infection depresses the rate of mitochondrial respiration in macrophages

Mitochondria are regarded as the energy factory of the cell that generates ATP through OXPHOS. It is reasonable to expect that on infection with *Mtb*, host energy metabolism is rewired, which has implications for understanding how *Mtb* causes disease. To examine the effect of mycobacterial infection on host OXPHOS, we made use of an extracellular flux analyzer (XF, Agilent Seahorse, Santa Clara, CA) and the mitochondrial respiration test (*Nicholls et al., 2010*) to determine key respiratory parameters in mammalian cells. These include basal respiration (Basal Resp), which is the initial OCR measured before addition of any inhibitors minus the non-mitochondrial respiration; ATP-linked respiration (ATP-linked OCR), which is determined after addition of oligomycin that inhibits ATP synthase (Complex V) and thus approximates the respiration required to drive ATP synthesis; proton leak, which is the difference between the ATP-linked OCR and the non-mitochondrial respiration; maximal respiration (Max Resp), which is induced after addition of carbonyl cyanide-4-(trifluoromethoxy)phenylhydrazone (FCCP) that uncouples ATP synthesis from electron transport; spare respiratory capacity (SRC), which is the difference between maximal respiration and basal respiration; and non-mitochondrial respiration (Non-Mito Resp), which is the OCR after addition of rotenone, a complex I inhibitor, and antimycin A, a complex III inhibitor (*Figure 1B*). SRC is an important parameter that reflects the ability of the cell to increase respiration to increase the supply of ATP in scenarios when the energy demand exceeds supply under conditions of stress or increased work load.

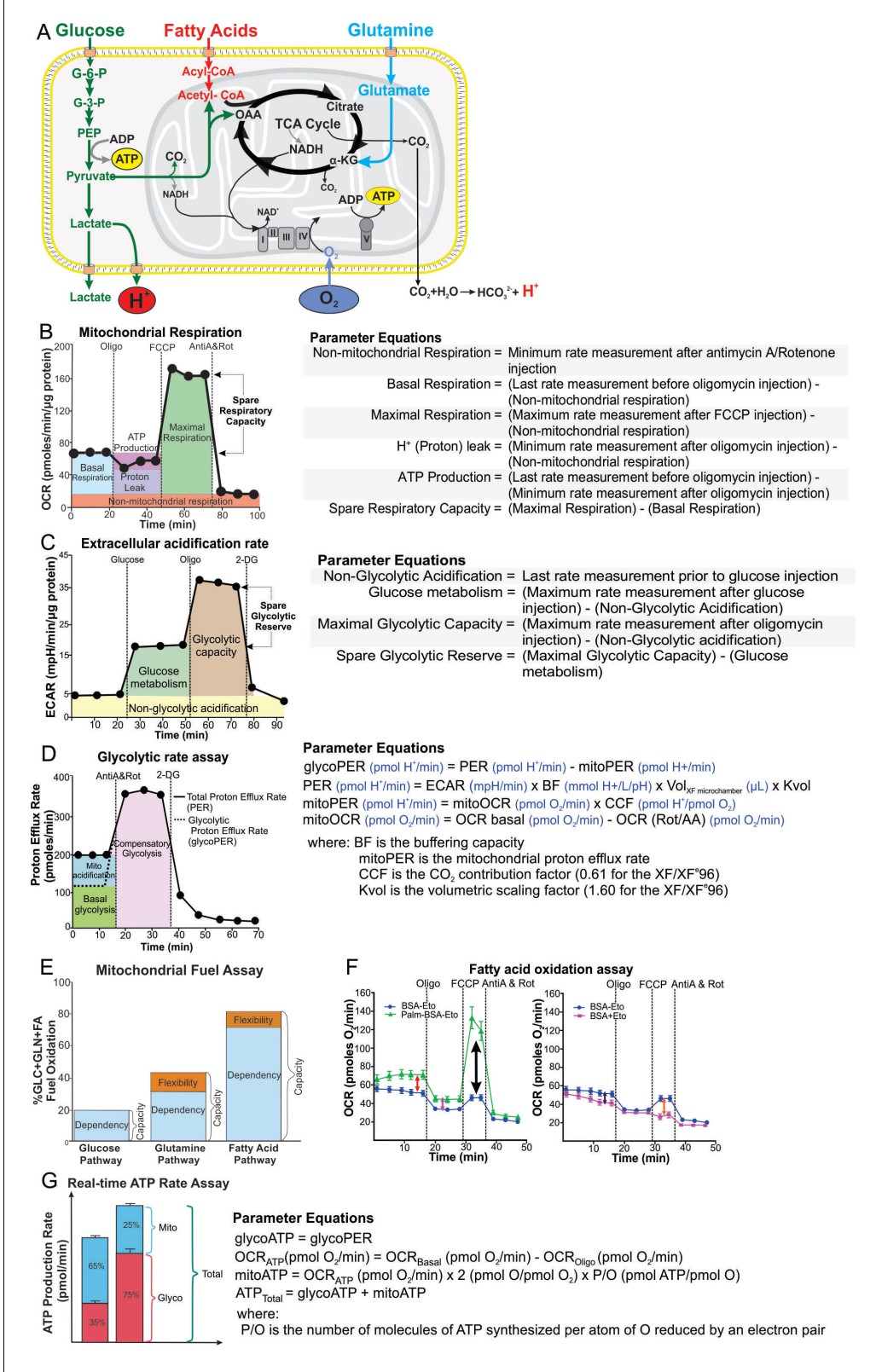

**Figure 1.** Schematic illustration of cellular metabolism pathways and XF assays used to analyze metabolic pathways. (**A**) The XF measures oxygen consumption rate (OCR) of the cell, which is mostly consumed at complex IV of the electron transport chain (ETC) in the mitochondria, and extracellular acidification rate (ECAR), which is generated from lactic acid produced from pyruvate, the end-product of glycolysis, and carbonic acid produced from $CO_2$ released during the TCA cycle. Assays performed on the XF include: (**B**) mitochondrial respiration test, (**C**) extracellular acidification test, (**D**)

*Figure 1 continued on next page*

*Figure 1 continued*

glycolytic rate assay, (E) mitochondrial fuel test, (F) fatty acid oxidation assay and (G) real-time ATP rate assay. Oligo, oligomycin; FCCP, cyanide-4-[trifluoromethoxy]phenylhydrazone; AntiA and Rot, antimycin A and rotenone; 2-DG, 2-Deoxyglucose; G-6-P, glucose-6-phosphate; G-3-P, glyceraldehyde-3-phosphate; PEP, phosphoenolpyruvate; α-KG, α-ketoglutarate; OAA, oxaloacetate.

DOI: https://doi.org/10.7554/eLife.39169.002

Several lines of evidence preclude any contribution of the infecting *Mtb* to the measured OCR of infected macrophages. Firstly, we have previously demonstrated that $10^6$ *Mtb* consumed 10–20 pmoles $O_2$/min (*Lamprecht et al., 2016*) in contrast to 100–200 pmoles $O_2$/min consumed by 80 000 to 100 000 macrophages under the basal conditions measured in this study. Thus, at a multiplicity of infection (MOI) of 1, $10^5$ *Mtb* would result in a negligible contribution (<1 pmoles $O_2$/min) to the OCR of the uninfected macrophages. Secondly, this basal OCR of *Mtb* was measured in media favourable for *Mtb* respiration and growth, whereas the intracellular macrophage environment is not conducive to 'healthy' *Mtb* respiration. Thirdly, *Mtb* infection at MOI of 1 and 2.5 progressively decreases the basal respiration (OCR) of both THP-1 macrophages and hMDMs (*Figure 2—figure supplement 1*) relative to that of uninfected macrophages. Should *Mtb* contribute to basal respiration (OCR), we should see an increase in OCR with increasing number of *Mtb* infecting the macrophages. Fourthly, the growth media of the infected macrophages after the time of infection and treatment was removed and the cells were washed in the XF assay medium before the XF assay, to remove most extracellular mycobacteria. When the washes of the macrophages were plated out on 7H11 agar plates, less than 200 CFU were obtained per well from the washes of the infected hMDMs (MOI 5), and less than 100 CFU per well from the infected THP-1 cells (MOI 5). To demonstrate that these extracellular mycobacteria do not contribute to the OCR readings of the infected macrophages, the final wash was transferred to a separate XF cell culture microplate and a separate mitochondrial respiration assay was performed on any extracellular bacteria present in the washes. The OCR and ECAR readings obtained were below 0 pmol/min and at 0 mpH/min, respectively, and the extracellular bacteria did not respond to the sequential injections of oligomycin, FCCP and rotenone and antimycin A (*Figure 2—figure supplement 2A–D*). Thus, the infecting mycobacteria do not contribute to the measured OCR of the infected macrophages under our conditions.

It is not possible to ensure that every cell will be infected in in vitro infections, thus the percentage of uninfected cells will contribute to the resulting XF profiles. We infected the macrophages with *Mtb*-green fluorescent protein reporter strain (*Mtb*-GFP) and used bright-field and fluorescence microscopy to determine the percentage of cells that were infected. We found that there was an increase in the percentage of infected cells with an increase in MOI of both the THP-1 cells and the hMDMs (*Figure 2—figure supplement 2E–G*). Although the percentage of uninfected cells will contribute to the readout of the XF profiles, previous studies have demonstrated that lipids shed by intracellular mycobacteria, such as TDM and PIM2, spread via the endocytic network throughout the macrophage, and via exocytic vesicles to neighboring uninfected cells (*Beatty et al., 2000*; *Xu et al., 1994*) and can elicit the production of proinflammatory cytokines (*Rhoades et al., 2003*). Consequently, the bioenergetic metabolism of the 'by-stander' uninfected cells will also be modulated. Thus, the XF profiles are providing collective data of a mixed population of macrophages.

Overall, our data demonstrated that strain pathogenicity and burden have distinct effects on virtually all respiratory parameters. *Figure 2A and B* shows that infection of THP-1 macrophages with *Mtb* or BCG significantly decreased the respiratory parameters: Basal Resp, ATP-linked OCR, proton leak and Max Resp (and SRC in the case of *Mtb*), and increased Non-Mito Resp. Similar patterns were observed at lower MOIs of 1 and 2.5, but to lesser degrees (*Figure 2—figure supplement 1A–D*). Infection with the dead *Mtb* only significantly reduced the respiratory parameters at MOIs of 2.5 and 5 (*Figure 2—figure supplement 1C–D* and *Figure 2A–B*).

*Mtb* infection of hMDMs dramatically reduced the respiratory parameters of the macrophage, while significantly increasing Non-Mito Resp (*Figure 2C–D*). Smaller reductions in the respiratory parameters were observed at lower *Mtb* MOIs of 1 and 2.5 with an increase in Non-Mito Resp (*Figure 2—figure supplement 1E–H*). Notably, contrary to *Mtb*, BCG infection increased the Max Resp and SRC of the hMDMs at all MOIs investigated. At a MOI of 5, BCG decreased the Basal Resp,

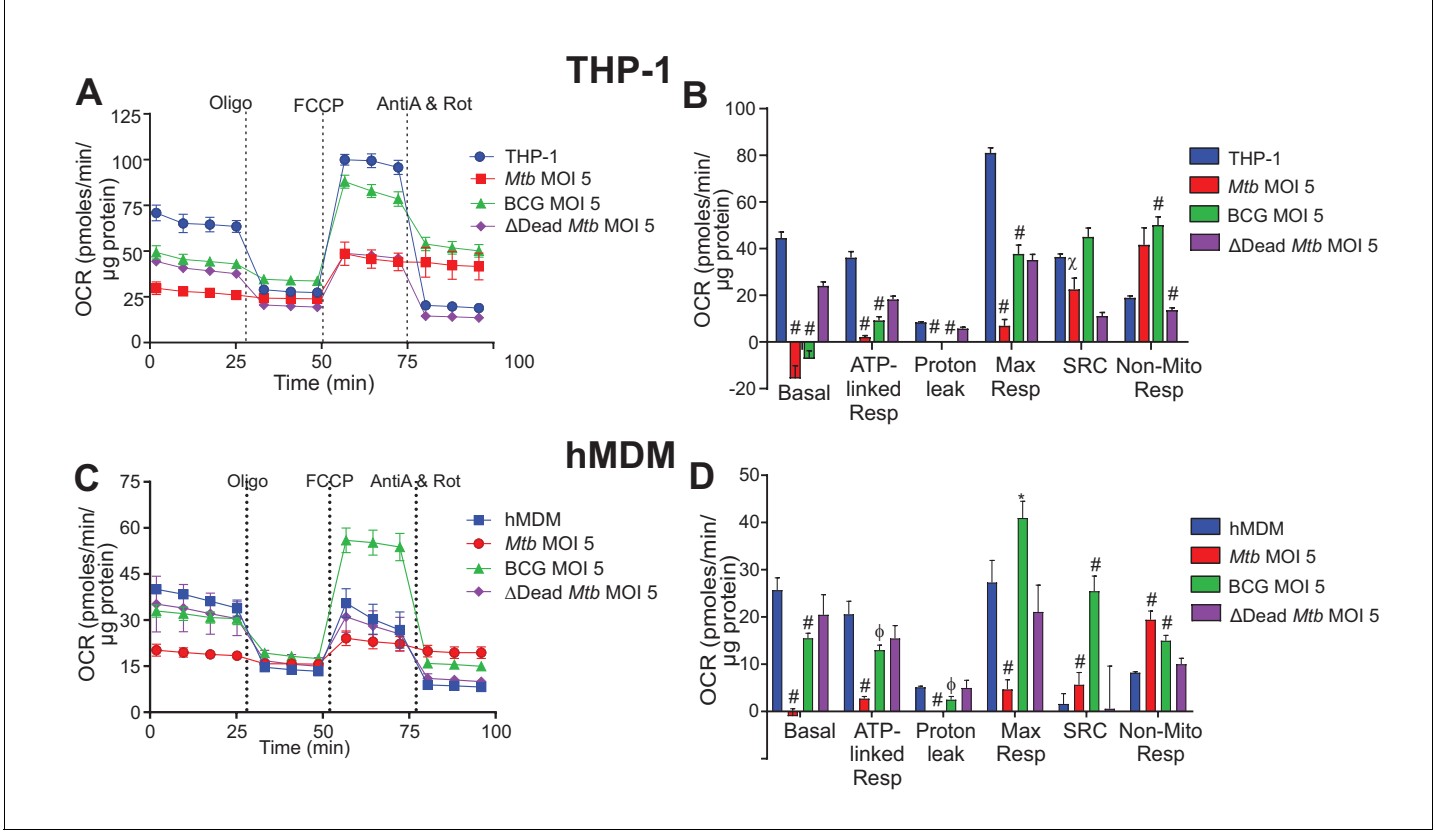

**Figure 2.** Respiratory profiles and parameters of infected macrophages are dependent on cell type, mycobacterial strain and MOI. Respiratory profiles (OCR) and respiratory parameters of (A–B) PMA differentiated THP-1 macrophages, and (C–D) hMDMs infected with *Mtb*, BCG and ΔDead *Mtb* (heat-killed *Mtb*) at MOIs of 5 for 24 h. Refer to *Figure 2—figure supplement 1* for profiles of lower MOIs. After obtaining basal respiration, cells were subjected to oligomycin (Oligo, 1.5 μM), which inhibits ATP synthase and demonstrates the mitochondrial ATP-linked OCR, followed by FCCP (cyanide-4-[trifluoromethoxy]phenylhydrazone), which uncouples mitochondrial respiration and maximizes OCR (1 μM for THP-1 and hMDMs), and finally antimycin A and rotenone (AntiA and Rot), which inhibit complex III and I in the ETC, respectively, and shut down respiration (0.5 μM of each for THP-1; 2.5 μM of each for hMDMs). Profiles and respiratory parameters are representative of three independent experiments. Data shown are the mean ± SD (n = 6 biological replicates). Student's t test relative to uninfected cells; #, $p < 0.0001$; χ, $p < 0.0005$; ϕ, $p < 0.001$; *$p < 0.005$; +, $p < 0.05$.
DOI: https://doi.org/10.7554/eLife.39169.003

The following figure supplements are available for figure 2:

**Figure supplement 1.** Respiratory profiles and parameters of infected macrophages are dependent on cell type, mycobacterial strain and MOI.
DOI: https://doi.org/10.7554/eLife.39169.004

**Figure supplement 2.** Contribution of extracellular mycobacteria to OCR and ECAR of infected macrophages and percentage of *Mtb*-infected macrophages.
DOI: https://doi.org/10.7554/eLife.39169.005

**Figure supplement 3.** Mitochondrial respiration assays without FCCP.
DOI: https://doi.org/10.7554/eLife.39169.006

ATP-linked OCR and proton leak, while increasing the Non-Mito Resp (*Figure 2C–D*). At lower MOIs, BCG had little effect on the other respiratory parameters (*Figure 2—figure supplement 1E–H*). Dead *Mtb* did not affect the respiratory parameters of the hMDMs at MOIs of 1 and 5, but a MOI of 2.5 increased the Max Resp and SRC of the macrophages as in the BCG infection (*Figure 2C–D* and *Figure 2—figure supplement 1E–H*).

In *Mtb*-infected macrophages, the oxidative burst (via NADPH oxidase, which consumes $O_2$) induced by a combination of infection, uncoupling with FCCP and inhibition of the ETC after treatment with antimycin A and rotenone increases the OCR above the initial OCR before treatment with oligomycin (*Figure 2A and C*). This results in the calculated basal respiration having a negative value (*Figure 2B and D*). As we have used standard equations to calculate the basal respiration (*Nicholls et al., 2010*), we propose that when the non-mitochondrial respiration is greater than the

initial OCR readings before the addition of any inhibitors, an additional mitochondrial respiration assay should be performed without the addition of the FCCP to determine the non-mitochondrial respiration, to obtain the values of basal respiration and proton leak. Using this format, the true (positive) values for basal respiration and proton leak are obtained (*Figure 2—figure supplement 3*). The basal respiration of both *Mtb* and BCG infections at a MOI of 2.5 and 5 were less than that of uninfected THP-1 cells, and the proton leak of BCG was less than that of the uninfected THP-1 cells (*Figure 2—figure supplement 3A–D*). The non-mitochondrial respiration was increased in both of these infections at a MOI of 5. Similar patterns were observed with the *Mtb*- and BCG-infected hMDMs at a MOI of 5 (*Figure 2—figure supplement 3E,F*).

In sum, there are profound contrasting respiratory differences among *Mtb,* BCG and dead *Mtb* infection of the macrophages. In particular, Max Resp, SRC and Non-Mito Resp are strongly influenced by the mycobacterial strain, burden and macrophage type. *Mtb* infection of hMDMs decreases Max Resp and SRC in contrast to BCG increasing Max Resp and SRC, and both strains increase Non-Mito Resp. SRC has consequences on how the macrophage responds to environmental stresses such as nutrient availability, redox state and changes in pH. Thus, an increase in the SRC of hMDMs following infection with potential vaccine candidates may aid identification of promising candidates. Strikingly, dead *Mtb* infection still alters the bioenergetic metabolism of the macrophage, in particular that of the THP-1 cells. This has implications for pharmacological killing of *Mtb*, as killing intracellular *Mtb* will not fully restore the macrophage's bioenergetic metabolism to that of the uninfected macrophage. However, pharmacological killing will improve the bioenergetic profile of the live *Mtb*-infected macrophages, in particular, the ATP-linked Resp and the Non-Mito Resp. Therefore, improvements in these parameters of the infected macrophages can be used as indicators of effective pharmacological killing of *Mtb* during screening of potential anti-TB drug leads in *Mtb*-infected macrophages.

## *Mtb* infection reduces the extracellular acidification rate of the macrophage

Glycolysis is the second pathway used to supply ATP for the energy requirements of the cell, in addition to anabolic intermediates. Here, we measured the glycolytic parameters of mycobacterial infected cells, including the glucose metabolism extracellular acidification rate after addition of glucose; the maximal glycolytic capacity (Gly capacity) following inhibition of OXPHOS ATP synthesis with oligomycin; and the non-glycolytic extracellular acidification measured after treatment of the mycobacterial infected cells with 2-deoxyglucose (2-DG), an inhibitor of hexokinase II, which catalyzes the first step of glycolysis (*Figure 1C*). The difference between the extracellular acidification rate of glucose metabolism rate and the maximal glycolytic capacity of the cells defines the spare glycolytic reserve.

*Mtb* strikingly decreased the glycolytic parameters of both types of macrophages after 24 h (*Figure 3*). In THP-1 cells, all the mycobacterial strains reduced the glycolytic parameters at MOIs of 5 (*Figure 3A–B*) and 2.5 (*Figure 3—figure supplement 1C,D*), with dead *Mtb* having the least effects. At a MOI of 1, *Mtb* decreased the glycolytic parameters, BCG increased the glycolytic parameters and dead *Mtb* had no effect (*Figure 3—figure supplement 1A–B*).

In hMDMs, *Mtb* at a MOI of 2.5 had little effect and at a MOI of 1 increased glucose metabolism extracellular acidification. (*Figure 3—figure supplement 1E–H*). Contrary to THP-1 cells, BCG and dead *Mtb* infection of hMDMs increased glucose metabolism acidification and the glycolytic capacity at all MOIs investigated. Increases in the non-glycolytic acidification were observed in the BCG and dead *Mtb* infections, probably as a result of the carbonic acid produced from $CO_2$ generated by the tricarboxylic acid cycle (TCA). These results underscore the different modulations of dead and live *Mtb* on macrophage bioenergetics.

In sum, marked glycolytic differences were observed between the virulent and non-virulent infections, with *Mtb* infection significantly reducing glucose metabolism extracellular acidification in the macrophages. BCG and dead-*Mtb* infections induced contrasting effects dependent on macrophage cell type, with a decrease in THP-1 glucose metabolism extracellular acidification versus an increase in hMDM extracellular acidification.

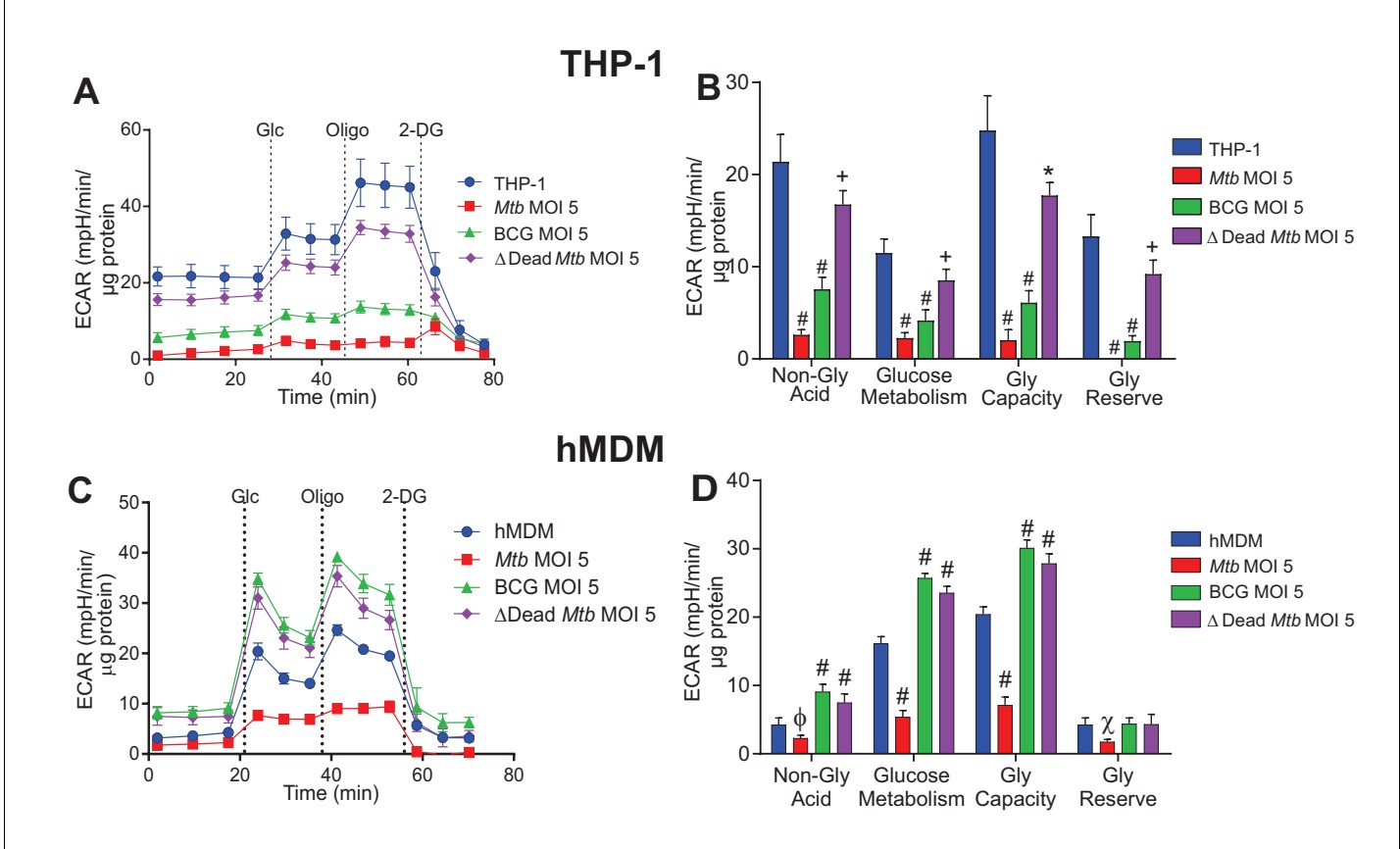

**Figure 3.** Extracellular acidification profiles and glycolytic parameters of THP-1 and hMDMs are affected by macrophage type, mycobacterial strain and MOI. ECAR profiles and glycolytic parameters of (A–B) PMA differentiated THP-1 macrophages, and (C–D) hMDMs infected with *Mtb*, BCG and dead *Mtb* at MOI of 5 for 24 h. Refer to *Figure 3—figure supplement 1* for profiles at lower MOIs. After obtaining non-glycolytic acidification, glucose (Glc, 10 mM) was added to the cells, followed by oligomycin (1.5 μM), which inhibits ATP synthase inducing maximal glycolysis to compensate for loss of mitochondrial generated ATP, and finally 2-deoxyglucose (2-DG, 100 mM) to inhibit glycolysis and demonstrate that the prior acidification was generated by glycolysis. Profiles and glycolytic parameters are representative of three independent experiments. Data shown are the mean ± SD (n = 6 biological replicates). Student's t test relative to uninfected cells; #, $p < 0.0001$; χ, $p < 0.0005$; ϕ, $p < 0.001$; *$p < 0.005$; +, $p < 0.05$.
DOI: https://doi.org/10.7554/eLife.39169.007

The following figure supplement is available for figure 3:

**Figure supplement 1.** Extracellular acidification profiles and glycolytic parameters of THP-1 and hMDMs are affected by macrophage type, mycobacterial strain and MOI.
DOI: https://doi.org/10.7554/eLife.39169.008

## *Mtb* infection shifts the bioenergetic phenotype of the macrophage towards quiescence

To determine how mycobacterial infection shifts the energy metabolism of the macrophage, basal OCR was plotted as a function of ECAR to form a bioenergetic phenogram that depicts the overall energy phenotypes of the macrophages. The energy phenotype of cells can be described as more aerobic, energetic, glycolytic or quiescent (*Figure 4*).

In THP-1 cells and hMDMs, infection with *Mtb* exhibited the most pronounced shift from an *Energetic* phenotype towards that of a *Quiescent* phenotype with increasing MOI (*Figure 4A and D*). In contrast, BCG and dead *Mtb* infections of THP-1 cells (MOIs 2.5 and 5) induced much smaller shifts towards quiescence (*Figure 4B and C*); whereas in hMDMs, only BCG infection at a MOI of 5 engendered a small shift towards quiescence. Dead *Mtb* did not affect the OCR of the hMDMs but decreased the ECAR at lower MOIs (*Figure 4F*), again underscoring the differences between live and dead *Mtb*. In sum, our data clearly demonstrate that *Mtb* infection shifts the energy phenotypes of human macrophages towards a metabolic quiescent state.

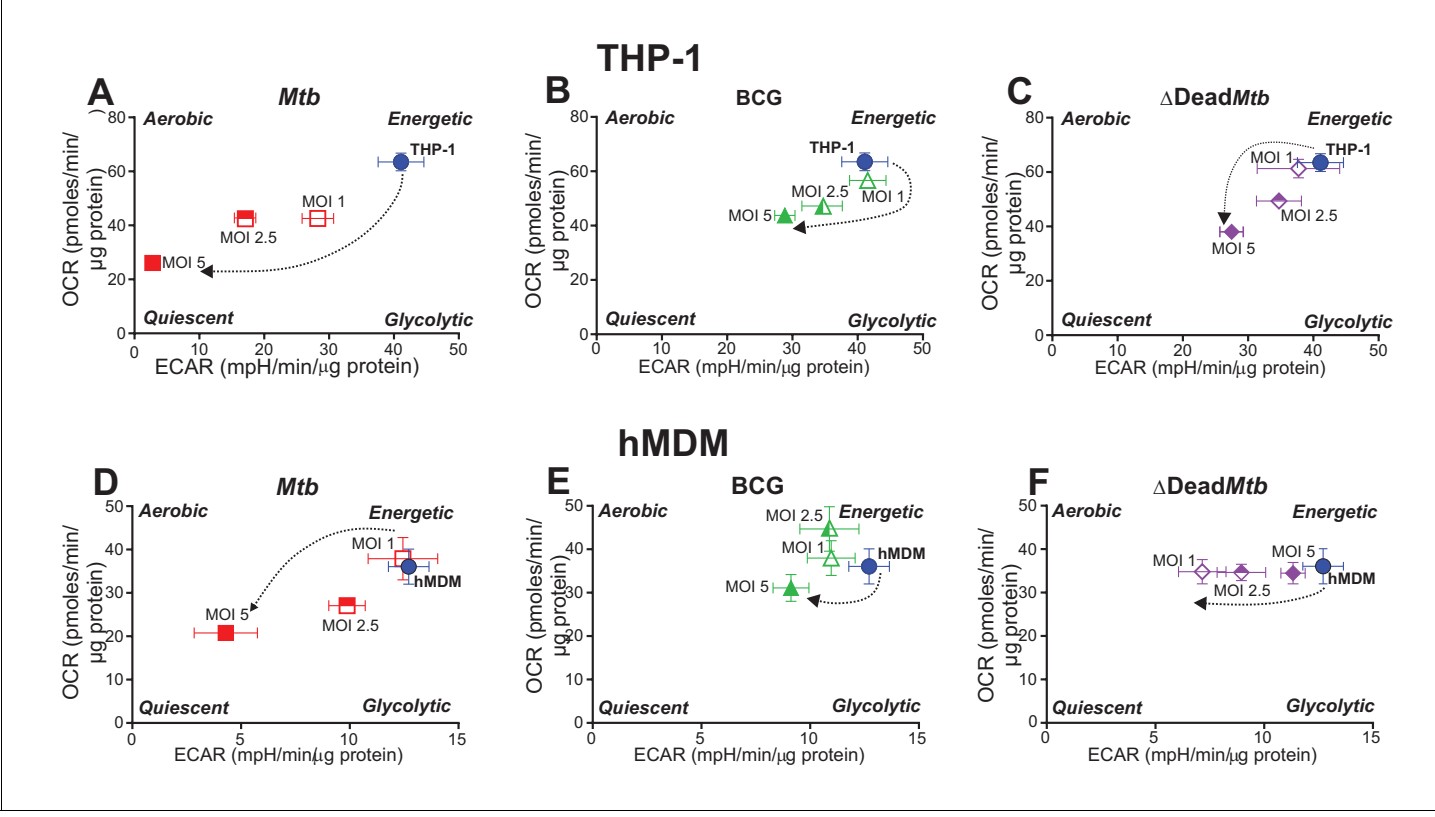

**Figure 4.** Phenograms demonstrate that increasing MOI of *Mtb* shifts macrophages towards quiescent energy phenotypes. Basal OCR and ECAR measurements from the respiratory assay (*Figure 2*) before addition of oligomycin were plotted to generate phenograms of (**A–C**) PMA-differentiated THP-1 cells and (**D–F**) hMDMs infected with *Mtb*, BCG and ΔDead *Mtb* at MOIs of 1, 2.5 and 5. Data are representative of three independent experiments. Data shown are the mean ± SD (n = 6 biological replicates). Student's t test relative to uninfected cells; #, $p < 0.0001$; χ, $p < 0.0005$; φ, $p < 0.001$; *$p < 0.005$; +, $p < 0.05$.

DOI: https://doi.org/10.7554/eLife.39169.009

### *Mtb* decreases the glycolytic proton efflux rate of macrophages

Previous studies using *Mtb* lysates (*Lachmandas et al., 2016*), irradiated killed *Mtb* and lactate measurements (*Gleeson et al., 2016*), or transcription profiling (*Shi et al., 2015*) led to the supposition that *Mtb* induces aerobic glycolysis for ATP generation, known as the Warburg effect. In XF analysis, bulk acidification of the extracellular medium, as measured by ECAR, is not specific for glycolysis as the mitochondrial TCA produces $CO_2$ that is partially hydrated in the extracellular medium and contributes to the acidification of the extracellular medium (*Mookerjee et al., 2015a*). In the glycolytic rate assay (*Mookerjee and Brand, 2015b*), inhibition of mitochondrial respiration after addition of rotenone and antimycin A enables calculation of the contribution of the mitochondrial respiration to the rate of proton efflux (*Figure 5A*). Subtraction of the mitochondrial proton efflux rate from the total proton efflux rate provides the glycolytic proton efflux rate (glycoPER) (*Figure 5— figure supplement 1A–D*). To confirm the specificity, 2-DG is added to inhibit glycolytic acidification (*Figure 5A*). Compensatory glycolysis refers to the ability of the cell to increase glycolysis after OXPHOS has been inhibited with rotenone and antimycin A.

The proton efflux rates illustrated in *Figure 5B–E* are the calculated values of the glycolytic PER without the acidification contribution from mitochondrial respiration. Basal glycolysis and compensatory glycolysis of the macrophages, which is induced when mitochondrial ATP synthase is inhibited thereby forcing the cell to use glycolysis to meet the cell's ATP requirements, are expressed as glycoPER (pmol/min/μg protein).

An immediate observation is that *Mtb* significantly reduces the basal and compensatory glycolytic rates of THP-1 cells (MOI 2.5 and 5, *Figure 5B–C*, *Figure 5—figure supplement 1E–H*). In contrast,

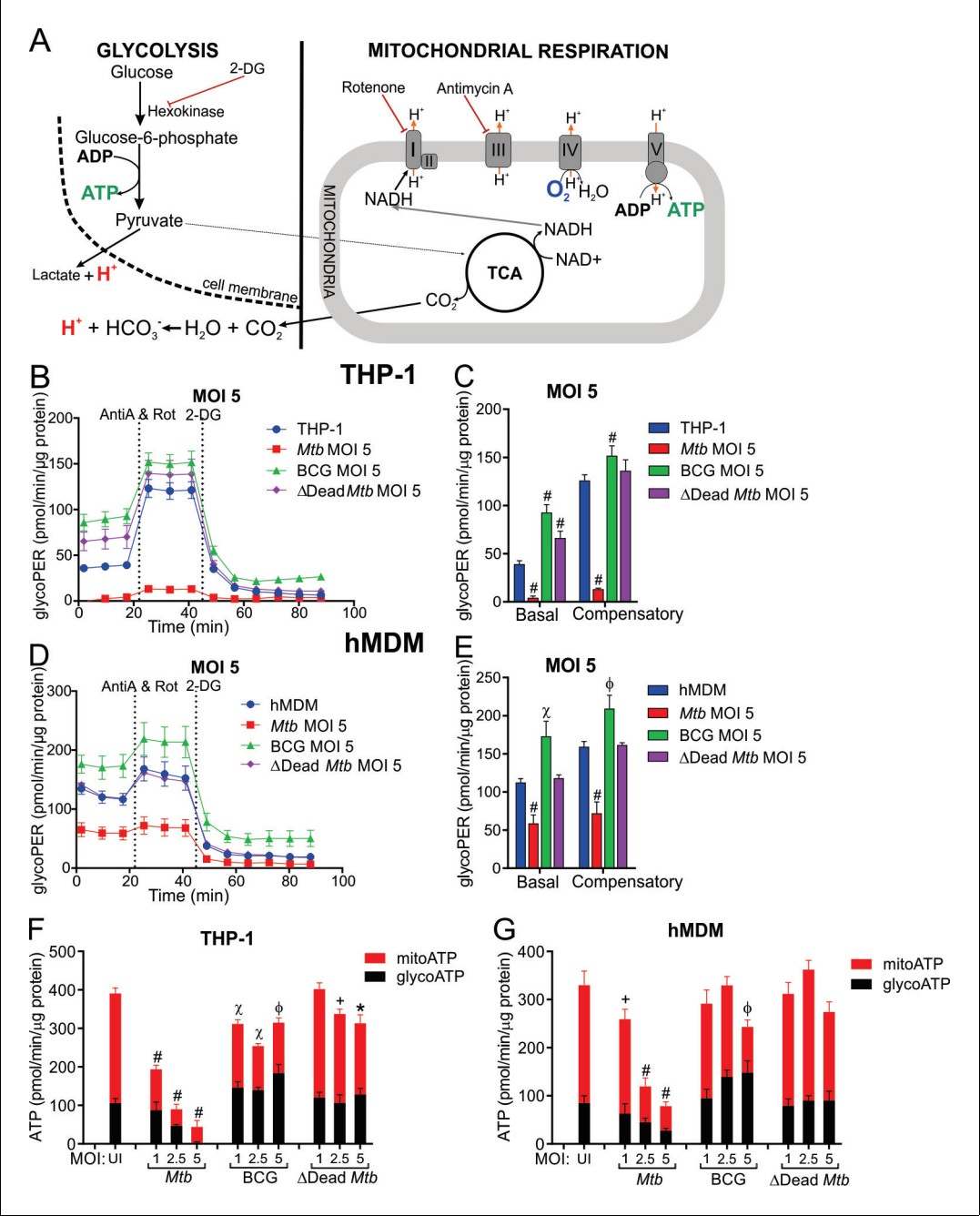

**Figure 5.** *Mtb* infection reduces the glycolytic proton efflux rate of macrophages. (**A**) Extracellular acidification can be caused by both lactate and protons produced from pyruvate, the final product of glycolysis, in addition to carbonic acid generated from $CO_2$ from pyruvate oxidation in the mitochondria. Calculating proton efflux rate (PER) enables the glycolytic PER to be elucidated separately from the mitochondrial PER (*Figure 5—figure supplement 1A–D*). (**B–E**) Basal and compensatory glycolytic PER of THP-1 cells (**B–C**) and hMDMs (**D–E**) infected with *Mtb*, BCG and ΔDead *Mtb* at MOI of 5 for 18 h. Refer to *Figure 5—figure supplement 1E–L* for profiles at lower MOIs. Following basal measurement of ECAR and OCR, to determine basal glycolytic PER, rotenone and antimycin A were added to determine compensatory PER. This was followed by addition of 2-DG to ensure that the PER observed was caused by glycolysis. Profiles and PER are representative of two independent experiments. Data shown are the mean ± SD (n = 6). Student's t test relative to the uninfected cells; #, $p < 0.0001$; χ, $p < 0.0005$; φ, $p < 0.001$; *$p < 0.005$; +, $p < 0.05$. (**F–G**) Total rate of ATP production was calculated as the sum of glycolytic ATP rate formation (equivalent to glycoPER) and mitochondrial-derived ATP rate formation that was estimated from the ATP-linked OCR, assuming a P/O ratio of 2.79. Rate of ATP formation in (**F**) THP-1 cells and (**G**) hMDM

*Figure 5 continued on next page*

*Figure 5 continued*

cells infected with *Mtb*, BCG or ΔDead *Mtb* at indicated MOI for 18 h. Refer to ***Figure 5—figure supplement 1M–N*** for % contribution of glycolysis and OXPHOS to the total rate of ATP production. Error bars are SD (n = 6 biological replicates). Student's t test; #, p < 0.0001; χ, p < 0.0005; ϕ, p < 0.001; *p < 0.005; +, p < 0.05.
DOI: https://doi.org/10.7554/eLife.39169.010
The following figure supplement is available for figure 5:

**Figure supplement 1.** *Mtb* infection reduces the glycolytic proton efflux rate of macrophages.
DOI: https://doi.org/10.7554/eLife.39169.011

BCG and dead *Mtb* infections increased the basal glycolytic rates of THP-1 cells at all MOIs examined. Furthermore, compensatory glycolysis was increased in BCG-infected cells at all MOIs, and in dead *Mtb* infections at a MOI of 1. Again, noticeable differences were observed between the live and dead *Mtb* infections.

In hMDMs, *Mtb* (MOI 5) decreased both the glycolytic rate and the compensatory glycolytic rate significantly (***Figure 5D–E***), but lower MOIs (1 and 2.5) did not affect the glycolytic rates of hMDMs (***Figure 5—figure supplement 1I–L***). BCG infection increased both the glycolytic rate and the compensatory glycolytic rates at all MOIs, whereas dead *Mtb* had no effect (***Figure 5D–E***, ***Figure 5—figure supplement 1I–L***).

In sum, profound differences were observed in the glycoPER of infections with virulent versus non-virulent mycobacterial strains. Noticeably, *Mtb* decreases the glycolytic PER in human macrophages at higher mycobacterial burdens, whereas BCG increases the glycolytic PER. However, dead *Mtb* infection increases the glycoPER in THP-1 cells and has no effect in hMDMs.

When we compared the glycoPER data, which is considered a more accurate measurement of the glycolytic rate (***Mookerjee and Brand, 2015b***), with ECAR data of the infected macrophages, we found that the observed discordances were dependent on the infecting mycobacterial strain. To illustrate, the reduced basal and compensatory glycoPER of *Mtb*-infected THP-1 cells (***Figure 5B–C***, ***Figure 5—figure supplement 1E–H***) resembled the observed decreased glycolytic ECAR and glycolytic capacity (***Figure 3A–B***, ***Figure 3—figure supplement 1A–D***). In contrast, distinct differences were observed in the BCG infection of the THP-1 cells, where significant increases in both basal and compensatory glycoPER (***Figure 5B–C***, ***Figure 5—figure supplement 1E–H***) were counter to significant reductions in the glucose metabolism ECAR and glycolytic capacity (***Figure 3A–B***, ***Figure 3—figure supplement 1C–D***). These opposing trends were not observed in the BCG infection of hMDMs, rather increased basal and compensatory glycoPER (***Figure 5D–E***, ***Figure 5—figure supplement 1I–L***) were reflected by increased glucose metabolism ECAR and glycolytic capacity (***Figure 3C–D***, ***Figure 3—figure supplement 1E–H***). However, conflicting observations between glycoPER and ECAR were again observed with dead *Mtb* infection of THP-1 cells with increased basal glycoPER (***Figure 5B–C***, ***Figure 5—figure supplement 1E–H***) compared to marginal or no reductions in glycolytic ECAR (***Figure 3A–B***, ***Figure 3—figure supplement 1A–D***).

Similarly, in hMDMs, the insignificant changes in basal and compensatory glycoPER in *Mtb* infection (MOIs 1 and 2.5) reflected the minimal changes in ECAR and glycolytic capacity (***Figure 5—figure supplement 1I–L***; ***Figure 3—figure supplement 1E–H***); and the decreased glycoPER at a *Mtb* MOI of 5 mirrored the decreased ECAR (***Figure 5D–E***, ***Figure 3C–D***). Likewise, BCG increased both the glycoPER and ECAR. In contrast, opposing trends were observed with dead *Mtb*, which had no effect on basal or compensatory glycoPER (***Figure 5D–E***, ***Figure 5—figure supplement 1I–L***), but increased glycolytic ECAR and glycolytic capacity (***Figure 3C–D***, ***Figure 3—figure supplement 1E–H***).

Overall, these differences reflect the accuracy of the glycoPER in measuring the rate of glycolytic acidification. The similar trends between glycoPER and ECAR observed in *Mtb* infections result from the suppression of OXPHOS that is more prominent in *Mtb* infection, thus having minimal effects on ECAR caused by pyruvate oxidation.

## *Mtb* reduces the rate of ATP production in macrophages

ATP is the energy currency of the cell and here, we investigate how *Mtb* infection modulates the rate of ATP production by the two ATP-producing pathways, glycolysis and OXPHOS, in the

macrophage. Intracellular ATP turnover can be quantitatively reflected by extracellular fluxes (*Mookerjee et al., 2017*). Subsequently, in the ATP production rate assay, we determined the total rate of ATP formation from the sum of the glycolytic rate of ATP formation, equivalent to glycoPER, and the mitochondrial-derived ATP rate of formation that was estimated from the ATP-coupled OCR assuming a maximum P/O ratio (mole of ATP generated per mole of [O] atoms consumed) of 2.79 (*Mookerjee et al., 2017*, *Figure 1G*).

*Mtb* infection significantly reduced the total ATP production rate in THP-1 cells (*Figure 5F*) and hMDMs (*Figure 5G*) inversely with an increase in MOI. This concurs with the quiescent energy phenotype observed in *Figure 4A and D*. However, notably in THP-1 cells, at lower *Mtb* MOIs of 1 and 2.5, the contribution of the glycolytic rate to total ATP production rate increased significantly by 18% and 26%, respectively, relative to uninfected THP-1 cells (*Figure 5—figure supplement 1M*). In contrast, only a moderate increase was observed at *Mtb* MOI of 2.5 in hMDMs (*Figure 5—figure supplement 1N*). BCG infection also reduced total ATP production rate in THP-1 cells at all MOIs relative to uninfected cells, but to a considerably lesser degree than *Mtb* (20% vs. 50% at a MOI of 1). In hMDMs, BCG only significantly decreased ATP production at a MOI of 5 by 26% (*Figure 5G*). Notably, the contribution of the glycolytic rate to total ATP production rate in the BCG-infected human macrophages increased with an increase in MOI (*Figure 5—figure supplement 1M–N*). Dead *Mtb* infection only reduced the total ATP production in THP-1 cells significantly at higher MOIs, having no significant effects on ATP production rates in hMDMs (*Figure 5F and G*).

In sum, *Mtb* significantly decreases the ATP production rate in human macrophages with an increase in MOI. Nevertheless, the contribution of glycolysis to the total ATP production rate was increased at lower MOIs in *Mtb*-infected THP-1 cells and in hMDMs at a MOI of 2.5. BCG infection progressively increased the contribution of glycolysis to the total ATP production rate with an increase in MOI in both THP-1 cells and hMDMs.

## *Mtb* infection decelerates bioenergetic metabolism in hMDMs and THP-1 cells

Analysis of metabolite levels using NMR or LC-MS/MS in conjunction with $^{13}$C-tracing is currently the gold standard for assessing metabolic states and fluxes. However, this method is invasive and only shows the metabolic status at a single point in time. XF analysis gives insight into the plasticity of the bioenergetic metabolism in real time in the presence of certain substrates, activators or inhibitors. To gain further insight into the XF data, we used $^{13}$C-tracing of metabolites extracted from uninfected and infected macrophages at the same time points of the XF analysis.

In *Mtb*-infected hMDMs, $^{13}$C-tracing indicated reduced total $^{13}$C-incorporation in the glycolytic intermediates: glucose-6-phosphate/fructose-6-phosphate (G6P/F6P), pyruvate and lactate; and reduced flux of incorporation into dihydroxyacetone phosphate/glyceraldehyde-3-phosphate (DHAP/G-3-P) and phosphoenolpyruvate (PEP) from that in uninfected, BCG and dead *Mtb*-infected hMDMs (*Figure 6A*). Reduced enrichment was also observed in the TCA metabolites (citrate, α-ketoglutarate (α-KG), fumarate and malate) of the *Mtb*-infected hMDMs (*Figure 6B*). This suggests decreased flux through glycolysis and the TCA cycle. In contrast, BCG and dead *Mtb* showed greater enrichment than the uninfected cells in most of the glycolytic intermediates, signifying increased flux through glycolysis in these infections. Different labelling patterns were observed in fructose-1,6-bisphosphate (*Figure 6A*), where *Mtb*-infected hMDMs had the greatest enrichment, possibly as a result of gluconeogenesis. Slower flux of incorporation was also observed in the pentose phosphate pathway (PPP) in *Mtb*-infected hMDMs (ribose-5-phosphate, sedoheptulose-7-phosphate and erythrose-4-phosphate, *Figure 6C*). The PPP is induced in inflammatory macrophages as it generates NADPH, which is required by NADPH oxidase to produce ROS (*Galván-Peña and O'Neill, 2014*). Similar to the hMDMs, in *Mtb*-infected THP-1 cells at a MOI of 2.5, reductions of enrichment in the glycolytic and TCA metabolites of *Mtb*-infected cells were significant with slower $^{13}$C incorporation in PEP and DHA/G3P (*Figure 6—figure supplement 1*) indicating a decelerated flux through glycolysis and the TCA cycle.

Increased secreted lactate is often used as a readout of increased glycolysis (*Gleeson et al., 2016*; *Braverman et al., 2016*). Our data demonstrate reduced $^{13}$C incorporation in the lactate and pyruvate in the supernatant fluid (SNF) of *Mtb*-infected hMDMs (MOI 5, *Figure 6D*) and in the lactate in the SNF of *Mtb*-infected THP-1 cells (MOI 2.5, *Figure 6E*). The total levels of normalised lactate (total peak area/3 × 10$^6$ cells, *Figure 6D and E*) in the SNF were significantly reduced in *Mtb*-

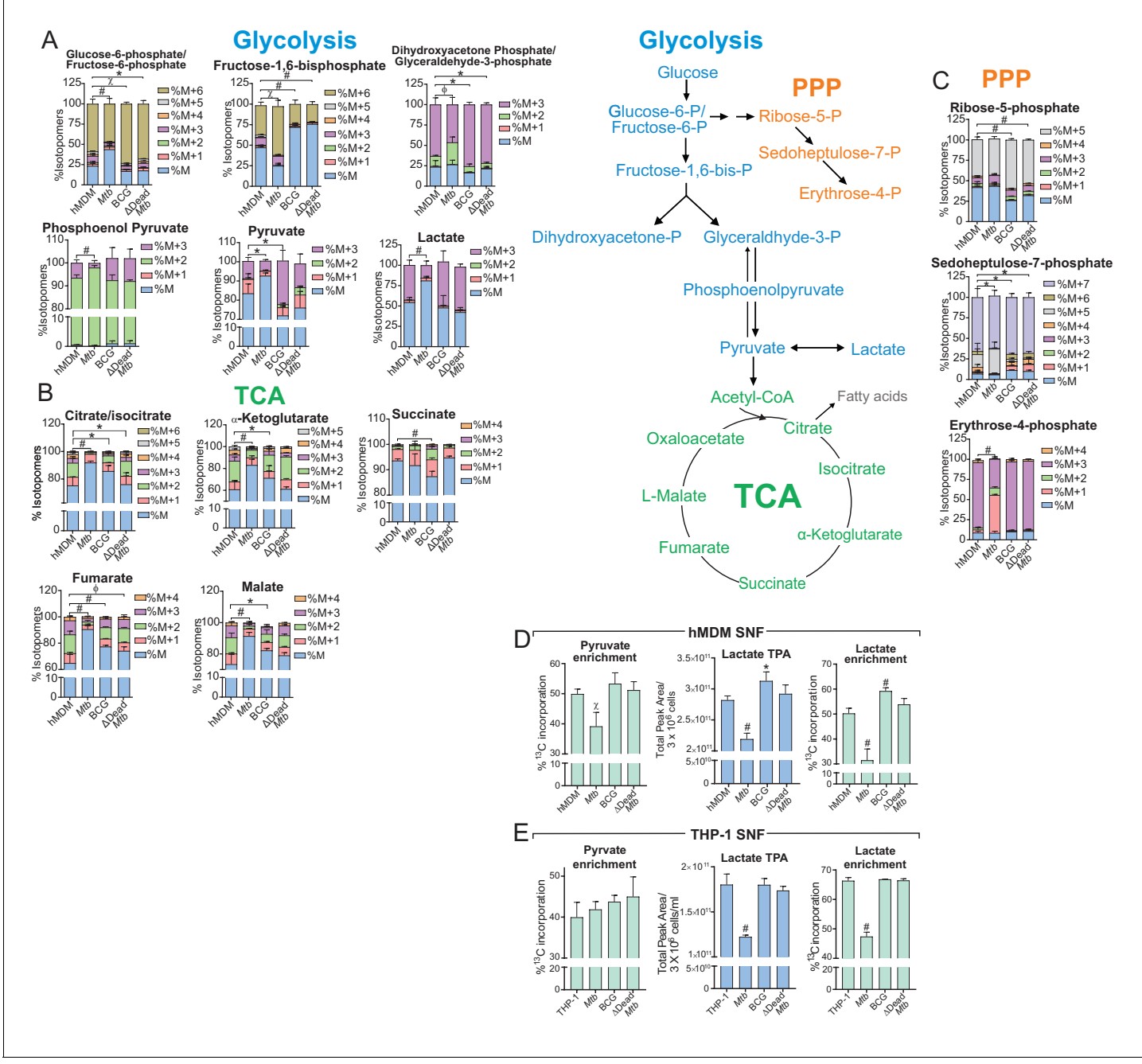

**Figure 6.** *Mtb* infection decelerates flux through glycolysis and the TCA cycle of the macrophage. (A–C) [13]C-tracing of metabolites extracted from hMDM cells infected with *Mtb*, BCG and ΔDead *Mtb* at MOI 5 (hMDM) for 10 h followed by incubation with [U-[13]C]glucose for 8 h. The stacked mass isotopomer distributions of the intracellular metabolites depict the contribution of glucose to (A) glycolysis, (B) the tricarboxylic acid (TCA) cycle and (C) the pentose phosphate pathway (PPP). *Figure 6—figure supplement 1* demonstrates the isotopomer distributions of the intracellular metabolites of THP-1 cells infected at a MOI of 2.5. (D–E) [13]C enrichment of pyruvate and lactate, including total peak areas of lactate in the supernatant of the (D) hMDM cells and (E) THP-1 cells used for metabolite analysis. Data are representative of two independent experiments and shown as the mean ± SD (n = 6 biological replicates). Student's t test relative to uninfected cells; #, $p < 0.0001$; χ, $p < 0.0005$; ϕ, $p < 0.001$; *$p < 0.005$; +, $p < 0.05$.
DOI: https://doi.org/10.7554/eLife.39169.012

The following figure supplement is available for figure 6:

**Figure supplement 1.** *Mtb* at a MOI of 2.5 decelerates flux through glycolysis in THP-1 cells and induces breaks in the TCA cycle at citrate and succinate.
DOI: https://doi.org/10.7554/eLife.39169.013

infected hMDMs and THP-1 cells, but significantly elevated in BCG-infected hMDMs. BCG- and dead *Mtb*-infected THP-1 cells had similar extracellular levels and enrichment to uninfected cells. This confirmed the reduced glycolytic rate (*Figure 3D*) and glycolytic PER (*Figure 5E*) observed in *Mtb*-infected hMDMs, the elevated glycolytic rates observed in BCG-infected hMDMs (*Figure 5E*) and the lower glycolytic PER observed in *Mtb*-infected THP-1 cells (*Figure 5—figure supplement 1H*).

In sum, the decreased isotope enrichment and flux observed in glycolytic and TCA metabolites of *Mtb*-infected human macrophages indicates a deceleration of central metabolism in these cells. This confirms the quiescent energy phenotypes observed in *Mtb*-infected macrophages in *Figure 4A and D*, the decreased glycolytic PER and ATP production rates observed in *Figure 5*, together with the decreased ECAR resulting from both glycolysis (lactate) as well as the carbonic acid generated from the carbon dioxide produced by the TCA cycle in *Figure 3*.

## *Mtb* increases the mitochondrial dependency of macrophages on fatty acids

The development of host-directed therapies targeting energy sensors in TB would be greatly assisted by the knowledge of substrate preferences of mitochondria in *Mtb*-infected macrophages and how these differ to those in infections with non-virulent strains and in uninfected macrophages. Usually, three substrates are utilized by mitochondria in living cells for OXPHOS to generate ATP, namely, glucose, glutamine (Gln) and fatty acids (*Figure 7A*). Here, we assessed the dependency, capacity and flexibility of the mitochondria to use metabolites generated from glucose, Gln or fatty acids in the mitochondrial fuel test (*Figure 1E*). Dependency is demonstrated when the mitochondria are unable to use other metabolites to compensate as fuel for OXPHOS when a certain metabolite pathway is blocked, for example, the glucose pathway when UK5099 inhibits the mitochondrial pyruvate carrier that transports pyruvate into the mitochondria. Flexibility is revealed when the mitochondria can compensate for the inhibited pathway by utilizing metabolites from the other pathways to fuel OXPHOS. The capacity refers to the overall ability of the mitochondria to use a metabolite as fuel for OXPHOS and is equivalent to the sum of the dependency and flexibility. In addition to UK5099, which blocks the glucose oxidation pathway; etomoxir, which inhibits carnitine palmitoyl-transferase 1A that transports long-chain fatty acids from the cytosol into the mitochondria, is used to examine inhibition of long-chain fatty acid oxidation; and bis-2-(5-phenylacetamido-1,3,4-thiadia-zol-2-yl)ethyl sulphide (BPTES), which is an allosteric inhibitor of glutaminase that converts Gln to glutamate that is further metabolised to α-KG, is used to investigate inhibition of the Gln oxidation pathway (*Figure 7A*).

We found minimal flexibility in the THP-1 cell line for all three substrates, regardless of infection (*Figure 7B*). *Mtb* infection increased THP-1 mitochondrial dependency on glucose by 18%, but BCG infection decreased glucose dependency by 14%, and dead *Mtb* infection did not affect the glucose dependency relative to uninfected cells (*Figure 7B*). Although *Mtb* infection induced a slight increase in Gln dependency of the THP-1 mitochondria, no changes were observed in Gln dependency in the BCG infection and dead *Mtb* infection (*Figure 7B*). THP-1 mitochondrial dependency on fatty acid oxidation increased by 14% in *Mtb* infection and decreased by 14% in dead *Mtb* infection relative to uninfected cells. Only dead *Mtb* increased the mitochondrial flexibility on long-chain fatty acid oxidation by 8%, which highlights the differences between infection with living or dead *Mtb*. The minimal changes observed in the mitochondrial fuel preferences of the THP-1 cells infected with BCG and dead *Mtb* align with the minimal changes observed in the phenograms of these infections at MOI of 1 (*Figure 4B and C*). Overall, the greater changes in the mitochondrial fuel dependencies of *Mtb*-infected THP-1 cells are reflected in the shift of *Mtb* infections away from an energetic to a quiescent energy phenotype (*Figure 4A*). Lastly, *Mtb* increases the fuel dependency and capacity of the mitochondria in THP-1 macrophages on glucose and long-chain fatty acids.

In contrast to THP-1 cells, major differences were observed in the mitochondrial fuel preferences of the infected hMDMs, which is supported by the following five lines of evidence. Firstly, *Mtb* infection of hMDMs decreased the mitochondrial dependency on glucose by 36% and slightly increased the glucose flexibility (7%), whereas BCG infection did not alter glucose dependency, but decreased the flexibility (by 68%) and capacity of the mitochondria to utilize glucose by 50% (*Figure 7C*). Secondly, dead *Mtb* infection increased the dependency of the hMDM mitochondria on glucose by 32% without any flexibility and decreased the glucose capacity by 35%. Thirdly, uninfected hMDM

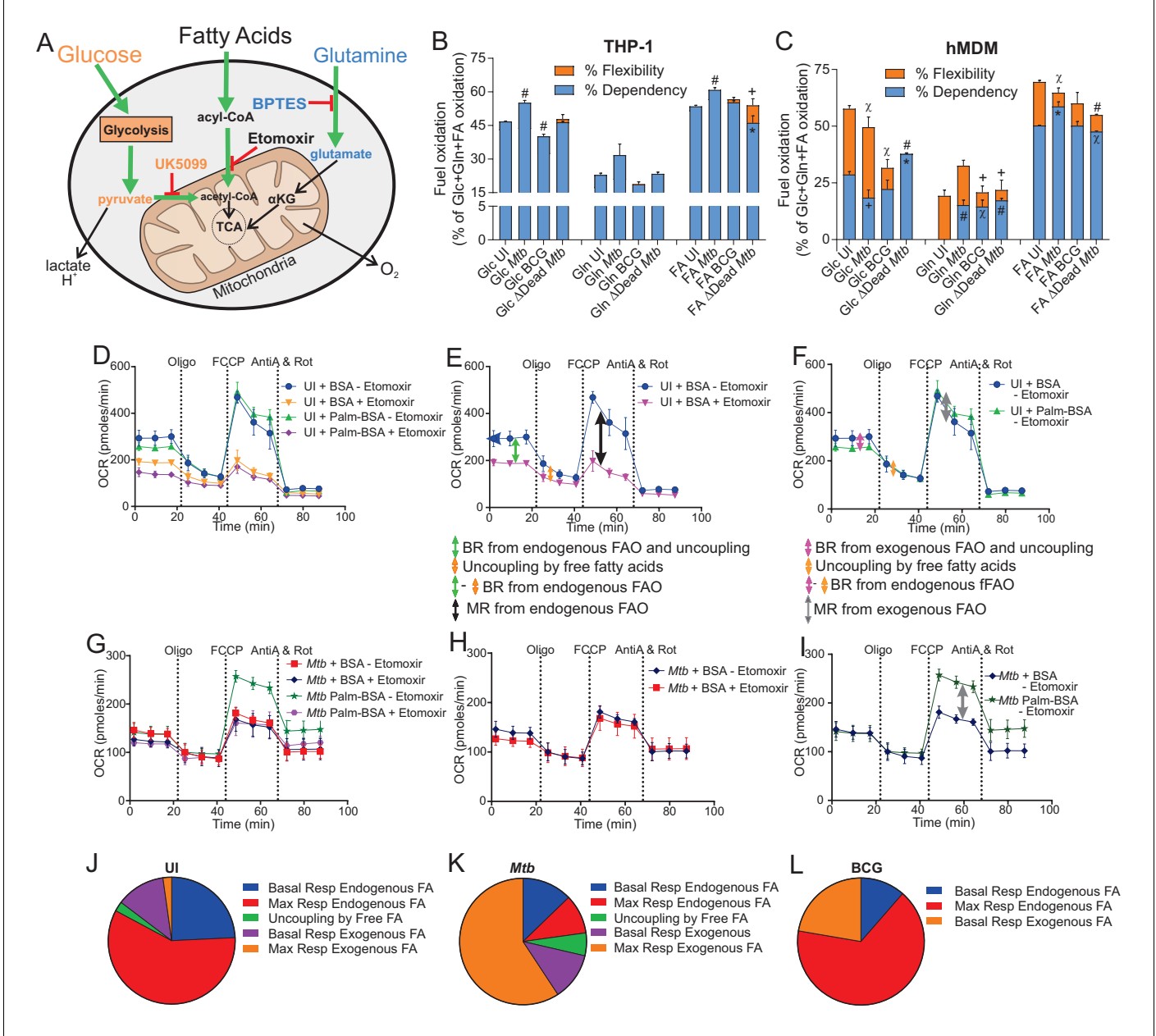

**Figure 7.** *Mtb* infection alters the mitochondrial substrate preference of macrophages. (A–C) UK5099, etomoxir and BPTES were used to assess the mitochondrial flexibility and dependency on glucose (Glc), fatty acids (FA) and glutamine (Gln) in (B) THP-1 and (C) hMDM cells infected with *Mtb*, BCG and ΔDead *Mtb* at MOIs of 1 and 5, respectively, for 18 h. Data shown are the mean ±SEM of five independent experiments. Student's t test relative to uninfected cells (UI); #, p < 0.0001; χ, p < 0.0005; φ, p < 0.001; *p < 0.005; +, p < 0.05. (D–I) The oxidation of endogenous or exogenous fatty acids in (D–F) uninfected hMDM cells in addition to (G–I) *Mtb*-infected macrophages at a MOI of 5 were assessed by adding a palmitate-BSA (palm-BSA) conjugate and BSA controls before analysis on the XF with the respiration test. Etomoxir was used to assess inhibition of the transport of long-chain fatty acids into the mitochondria. BR, basal respiration; MR, maximal respiration (which gives a measure of respiration under conditions of stress); FAO, fatty acid oxidation; FA, fatty acids. (J–L) Pie charts illustrating the mitochondrial fatty acid preferences of (J) uninfected, (K) *Mtb*- and (L) BCG-infected hMDMs under basal and stressful conditions. Profiles and pie charts are representative of two independent experiments (n = 6 biological replicates).
DOI: https://doi.org/10.7554/eLife.39169.014

demonstrated no dependency on Gln oxidation, whereas all the infections generated a dependency on Gln to similar levels ($\approx$ 15%). Fourthly, both BCG and dead *Mtb* infection decreased the flexibility of the mitochondria for Gln, whereas *Mtb* infection increased the capacity of hMDM mitochondria to utilize Gln by 68%. Fifthly, long-chain fatty acid dependency was increased in *Mtb* infection by 14% and marginally decreased in dead *Mtb* infection by 5%. Both infections decreased the flexibility for fatty acids and the mitochondrial capacity to use fatty acids in dead *Mtb* infections was reduced by 21%.

In summary, we have shown that all the infections induced hMDM dependency for Gln, *Mtb* decreased hMDM dependency for glucose and increased hMDM dependency for long-chain fatty acids. BCG infection had minimal effects on glucose and long-chain fatty acid oxidation, and dead *Mtb* increased hMDM dependency for glucose with no flexibility.

## *Mtb* switches the dependency of hMDMs from endogenous fatty acids to exogenous fatty acids

After observing that *Mtb* infection increased the hMDM mitochondrial dependency on fatty acids, we next examined the source of these fatty acids in infected hMDMs under basal and stressed conditions. Using the mitochondrial stress test, etomoxir, palmitate conjugated to bovine serum albumin as a fatty acid substrate and bovine serum albumin (BSA) as a control, the fatty acid oxidation assay measured the intrinsic rate and capacity of a cell to oxidize exogenously added fatty acids (the palmitate-BSA conjugate, *Figure 1F*). As this is influenced by the availability of other substrates, such as glucose, and stores of endogenous substrates such as glycogen, triglycerides and amino acids, the cells were cultured overnight in minimal media. Furthermore, the oxidation of endogenous fatty acids was measured using the BSA control and the mild uncoupling of the mitochondria induced by free fatty acids present in the BSA and palmitate-BSA conjugate preparation was also deduced.

Our profiles of BSA ± etomoxir clearly demonstrate that uninfected hMDMs utilize endogenous fatty acid oxidation under stressful conditions that increase energy demands, but the *Mtb*-infected hMDMs do not. *Figure 7D* shows the respiration profiles of the uninfected hMDMs with the BSA control or the palmitate-BSA conjugate both with and without etomoxir; *Figure 7G* illustrates the same profiles of *Mtb*-infected hMDMs. When the profiles of the BSA controls ± etomoxir are analyzed in *Figure 7E and H*, the difference between the basal respirations gives the respiration casued by endogenous fatty acid oxidation in addition to the OCR resulting from the uncoupling caused by free fatty acids (green arrow, *Figure 7E*). The difference between the ATP-linked respirations (orange arrow, after the addition of oligomycin, *Figure 7E and F*) is equivalent to the OCR induced by the free fatty acid uncoupling. Subtraction of the OCR from fatty acid coupling from the difference between the basal respirations of the two profiles gives the OCR resulting from endogenous fatty acid oxidation. Hence, uninfected hMDMs (*Figure 7D*) utilize endogenous fatty acids to a greater degree than do *Mtb*-infected hMDMs for basal respiration (*Figure 7H*). *Figure 7E* shows that after the addition of FCCP, which is added to examine the response under stress, the large difference between the two profiles generated with BSA ± etomoxir (black arrow, *Figure 7E*) indicates that the uninfected hMDMs utilize endogenous fatty acid oxidation under stressed energy demands, whereas *Mtb*-infected macrophages do not (*Figure 7H*).

Our profiles of the palmitate-BSA conjugate and that of BSA alone show that the *Mtb*-infected hMDMs rely on fatty acid oxidation of exogenous fatty acids under stressful conditions. When the profile of the palmitate-BSA conjugate without etomoxir is compared to that of the BSA control without etomoxir, the basal respiration (*Figure 7F*, pink arrow) and respiration under stressed conditions (*Figure 7F*, grey arrow) resulting from exogenous fatty acid oxidation can be measured using the same differences as described for the endogenous fatty acid oxidation. Thus, *Figure 7F* indicates that uninfected hMDMs do not utilize exogenous fatty acid oxidation to meet their energy requirements under basal or stressed energy demands. In contrast, *Mtb*-infected hMDMs depend on the oxidation of exogenous fatty acids under stressed conditions (*Figure 7I*).

The pie charts summarise the findings of our fatty acid oxidation assays on the hMDMs. Uninfected hMDMs depend on the oxidation of endogenous fatty acids, especially under stressed conditions (*Figure 7J*). *Mtb*-infected hMDMs rely on the oxidation of exogenous fatty acids under conditions of stress and to a much lesser extent on oxidation of the endogenous fatty acids (*Figure 7K*). BCG-infected hMDMs utilized endogenous fatty acid oxidation to meet stressed energy

demands, but utilized both exogenous fatty acid oxidation and endogenous fatty acid oxidation for basal energy demands (*Figure 7L*).

Thus, the fatty acid oxidation data enable identification of the source of fatty acids used by hMDMs under different energetic demands demonstrating that *Mtb* infection induces a shift in the macrophage fatty acid oxidation from the utilization of endogenous to exogenous fatty acids. This could be explained by *Mtb* dysregulating host lipid metabolism to sequester triacylglycerol into lipid bodies in the macrophage (*Daniel et al., 2011*; *Russell et al., 2009*). Hence, *Mtb*-infected macrophages rely on exogenously added fatty acids for mitochondrial fuel under stressful conditions.

## Discussion

Although nearly all immune cells exhibit a degree of metabolic flexibility, the mechanisms whereby pathogenic intracellular bacteria trigger shifts in bioenergetic metabolism of the host immune cells to survive during infection are largely unexplored. Here we have established a non-invasive, high resolution quantitative approach for studying bioenergetic metabolism of human macrophages infected with live virulent pathogenic and non-pathogenic mycobacteria. The most important conclusions from the current study are that unlike non-pathogenic or dead mycobacteria, *Mtb* uniquely decelerates both glycolysis and OXPHOS to enter a state of metabolic quiescence and consequently decreases the rate of ATP production of the macrophage. We also showed that *Mtb* infection reduced mitochondrial dependency on glucose, enhanced the mitochondrial dependency on fatty acids and shifted mitochondrial dependency from endogenous fatty acids in uninfected macrophages to exogenous fatty acids for survival under conditions of stress. The differences in several quantitative bioenergetic parameters between the infections of pathogenic and non-pathogenic mycobacteria, in addition to the differences detected between infections with living and dead *Mtb*, suggest potential new strategies for evaluating vaccine or drug candidates. Collectively, this experimental evidence identified glycolysis and endogenous fatty acid metabolism as potential targets for metabolic restoration via host-directed therapeutic intervention.

The findings in this study have broad implications for understanding the metabolic status of an infected cell, how it can be measured, and how bacterial pathogens reprogram the host metabolism to initiate and maintain disease. It is difficult to define what constitutes the metabolic health of an immune cell because of the metabolic plasticity necessary to fulfil their effector functions (*Loftus and Finlay, 2016*). Their functions are determined by the site of infection, the pathogen present, the extent of pathogenesis, and the redox and metabolic state of the microenvironment. Furthermore, virulence factors unique to each pathogen invariably alter the metabolism of immune cells to enhance survival of the pathogen and subvert adaptive immune responses. Hence, the metabolic health of an immune cell is likely to be defined by its ability to execute metabolic plasticity to fulfil its effector functions to protect the host. Given this, pathogens prove to be successful in establishing infection by inhibiting the metabolic plasticity of the immune cells. Here, we demonstrate for the first time how quantifiable bioenergetic parameters of the host, including the respiratory parameters, especially spare respiratory capacity; basal and compensatory glycolytic proton efflux rates; mitochondrial and glycolytic ATP production rates; in addition to the mitochondrial capacity of the host to oxidize glucose, glutamine or fatty acids, can be used to accurately measure and track disease established as a result of intracellular pathogens reprogramming host metabolism. These measurable parameters can be similarly applied to study host reprogramming induced by other intracellular pathogens such as *Brucella abortus*, *Francisella tularensis*, *Listeria monocytogenes*, *Salmonella typhi* and *Neisseria meningitidis.* Furthermore, this platform will now open new avenues to rapidly and quantifiably assess drug and vaccine efficacy to reverse the pathogen-induced rewiring of host metabolism. These findings encourage further exploitation of this platform to measure the bioenergetic modulations of infection on polarized macrophages and other cell populations ex vivo following host-directed treatments of the TB mouse model.

A robust glycolytic response is considered a hallmark of positive metabolic change in most activated immune cells and is essential for sufficient ATP generation and anabolism of biosynthetic intermediates. However, live *Mtb* decelerates glycolysis in infected macrophages as evidenced by the reduced glycolytic proton efflux rates of *Mtb*-infected hMDMs and THP-1 macrophages. Conversely, BCG infection increases the glycolytic proton efflux rate of the macrophages as expected and increases the glycolytic contribution to the total rate of ATP production. Thus, the shift in the

bioenergetic profiles of BCG-infected macrophages from uninfected macrophages demonstrates an appropriate healthy metabolic response to eradicate an infection. The reduced glycolytic rate in *Mtb*-infected macrophages points towards mechanisms of *Mtb* to subvert apt innate immune responses that would translate into suitable activation of the adaptive immune response.

This raises the important question: How does *Mtb* decelerate glycolysis? A plausible candidate is citrate as high levels of this metabolite allosterically inhibit phosphofructokinase (*Usenik and Legiša, 2010*). Phosphofructokinase catalyzes the conversion of fructose-6-phosphate and ATP to fructose-1,6-biphosphate and ADP. This reaction is one of the key regulatory and rate-limiting steps of glycolysis and inhibition of this reaction downregulates glycolysis. Indeed, our $^{13}$C tracing data demonstrates a reduced rate of glycolysis and high levels of citrate in the *Mtb*-infected hMDMs, pointing to phosphofructokinase as a target for pharmacological upregulation.

Secreted lactate is often used as a measure of glycolysis and increased lactate secretion has been used by researchers in the TB field to illustrate the switch to aerobic glycolysis in *Mtb*-infected macrophages (*Gleeson et al., 2016*; *Lachmandas et al., 2016*; *Braverman et al., 2016*). However, under our conditions, *Mtb* infection of hMDMs and THP-1 cells reduced lactate secretion, which was measured using stable isotopes and LC-MS/MS. Some of these differences can be attributed to our use of human monocyte-derived macrophages versus their use of mouse bone marrow-derived macrophages (*Braverman et al., 2016*), the use of stimulation with *Mtb* lysate (*Lachmandas et al., 2016*) or infection with γ-irradiated *Mtb* (*Gleeson et al., 2016*) rather than infection with live *Mtb* in addition to different protocols to generate hMDMs. However, our results do challenge current dogma that *Mtb* infection induces the Warburg effect in macrophages.

Ratios of OCR to ECAR have also been used to illustrate a switch from OXPHOS to glycolysis (*Gleeson et al., 2016*); however, ECAR is a measure of the extracellular acidification resulting from both glycolysis and oxidation of pyruvate in the TCA cycle (*Mookerjee et al., 2015a*). Thus, OCR/ECAR ratios (or ECAR/OCR) are not representative of the ratio of the rate of OXPHOS to the rate of glycolysis, although they are often interpreted as such (*Gleeson et al., 2016*; *Lachmandas et al., 2016*). Nevertheless, measurements of the total proton efflux rate followed by inhibition of the mitochondrial ETC to measure the mitochondrial contribution to the total proton efflux rate enables calculation of the glycolytic PER (*Mookerjee and Brand, 2015b*). Using the proton efflux rate, we found that *Mtb* reduced the glycoPER in human macrophages in contrast to BCG that increased the glycoPER, which is the expected response from inflammatory macrophages (*O'Neill and Pearce, 2016*).

In sum, the energy phenotype reminiscent of quiescence that we observed in *Mtb*-infected THP-1 macrophages and hMDMs may prove to be favourable for the development and maintenance of *Mtb* persistence. These findings suggest that pharmacological enhancement of glycolysis in the *Mtb*-infected macrophage may initiate and improve the anti-mycobacterial properties of the macrophage.

Metabolic plasticity of the immune cells entails changing substrates for bioenergetic metabolism to perform effector functions. Pathogens often establish successful infections by altering the substrate preferences of the immune cells to create a niche promoting the pathogen's survival and proliferation. Monitoring mitochondrial oxygen consumption in the presence of modulators affecting substrate utilisation or availability gives insight into the pathogen tempering of substrate preferences of the infected cells. For example, we found that infection of the hMDMs generates a dependency on Gln as opposed to no Gln dependency in uninfected cells, and infected THP-1 cells increase their dependency on Gln. This suggests the anapleurotic replenishment of α-KG as a result of citrate being used in other processes, such as fatty acid synthesis (*Mehrotra et al., 2014*). The decreased dependency on glucose observed in *Mtb*-infected hMDMs correlates with the reduced glycolytic rate detected in these macrophages. In contrast, the increased mitochondrial dependency of *Mtb*-infected hMDMs on fatty acid oxidation may result from *Mtb*-mediated induction of foamy macrophages. Foamy macrophages are formed through accumulation of lipid bodies composed of triglycerides and cholesteryl esters (*Thiam et al., 2013*), which are produced from malonyl Co-A and mevalonate, respectively. In turn, these substrates are produced from catabolism of citrate that is formed from the condensation of oxaloacetate and acetyl-CoA. *Singh et al. (2012)* found that virulent *Mtb* strains induced a pronounced increase in the synthesis of acetyl-CoA. A potential source of increased acetyl-CoA synthesis is macrophage fatty acid oxidation, which would substantiate the increased mitochondrial dependency of *Mtb*-infected macrophages on fatty acids. The observed

switch from dependence on endogenous to exogenous fatty acids induced by *Mtb* infection is likely a result of 3-hydroxybutyrate inhibiting endogenous lipolysis to promote lipid body formation (*Singh et al., 2012*), forcing the macrophage to use exogenous fatty acids to maintain viability and produce acetyl Co-A for the ultimate formation of triglycerides and cholesteryl esters. Thus, the reduced dependency on glucose and the increased dependency on fatty acids identify glycolysis and fatty acid oxidation as potential targets for host-directed therapies

In conclusion, we identified new paradigms in TB host metabolism, including *Mtb* decelerating OXPHOS and glycolysis in the human macrophages resulting in decreased ATP production. The decelerated bioenergetics decreased the dependency of the macrophage on glucose with a compensating increased capacity for glutamine and dependency on fatty acids, of which *Mtb* induces a shift from oxidation of endogenous fatty acids to that of exogenous fatty acids. Our findings suggest that restoration of glycolysis and endogenous fatty acid oxidation would augment the development of inflammatory effector functions in the *Mtb*-infected macrophage. In contrast, infection with the vaccine strain BCG increases glycolysis of human macrophages, characteristic of a healthy inflammatory response, in addition to increasing the macrophage's spare respiratory capacity. Thus, the quantifiable respiratory and glycolytic parameters established in this study would enable direct measures in the assessment of the efficacy of future anti-TB drug leads and vaccine candidates.

# Materials and methods

**Key resources table**

| Reagent type (species) or resource | Designation | Source or reference | Identifiers | Additional information |
|---|---|---|---|---|
| Strain, strain background (*Mycobacterium Tuberculosis*) | *Mtb* | BEI resources | NR-123 | |
| Strain, strain background (*Mycobacterium bovis*) | BCG | Stratens Serum Institut | | Danish BCG vaccine strain 1331 |
| Strain, strain background (*Mycobacterium Tuberculosis*) H37Rv AYs330 | *Mtb*-GFP | Steyn Laboratory | | GFP Reporter strain |
| Cell line (*Homo sapiens*) | THP-1 | ATCC Cat# TIB-202 | RRID:CVCL_0006 | |
| Biological sample (*Homo sapiens*) | Buffy coats | South African National Blood Service | | |
| Biological sample (*Homo sapiens*) | Human plasma | South African National Blood Service | | |
| Antibody | CD14 microbeads, Human | Miltenyi Biotec | Cat# 130-050-201 | MACS (1:25) |
| Commercial assay or kit | Seahorse XF Mito Fuel Flex Test Kit | Agilent | Cat# 103260–100 | |
| Commercial assay or kit | Lookout Mycoplasma PCR detection kit | Sigma | Cat# MO0035 | |
| Chemical compound, drug | Seahorse XF Palmitate-BSA FAO Substrate | Agilent | Cat# 102720–100 | |

*Continued on next page*

*Continued*

| Reagent type (species) or resource | Designation | Source or reference | Identifiers | Additional information |
|---|---|---|---|---|
| Chemical compound, drug | Oligomycin (from *Streptomyces diastatoc hromogenes*) | Sigma | Cat# O4876 | |
| Chemical compound, drug | Carbonilcyanide p-triflourometh oxyphenylh ydrazone (FCCP) | Sigma | Cat# C2920 | |
| Chemical compound, drug | Antimycin A | Sigma | Cat# A8674 | |
| Chemical compound, drug | Rotenone | Sigma | Cat# R8875 | |
| Chemical compound, drug | D-(+)-Glucose | Sigma | G8270 | |
| Chemical compound, drug | GlutaMAX™ | Gibco | Cat# 35050–038 | |
| Chemical compound, drug | DMEM | Lonza | Cat# BE12-604F | |
| Chemical compound, drug | L-Carnitine hydrochloride | Sigma | Cat# C0283 | |
| Chemical compound, drug | NaCl | Sigma | Cat# S3014 | |
| Chemical compound, drug | KCl | Sigma | Cat# SAAR50 42020EM | |
| Chemical compound, drug | CaCl$_2$ | Sigma | Cat# 1023780500 | |
| Chemical compound, drug | HEPES | Sigma | Cat# H0887 | |
| Chemical compound, drug | Sodium pyruvate | Sigma | Cat# S8636 | |
| Chemical compound, drug | XF Base medium | Agilent | Cat# 102353–100 | |
| Chemical compound, drug | Etomoxir | Sigma | Cat# E1905 | |
| Chemical compound, drug | Formalin Buffered, Neutral | Sigma | Cat# SAAR24 36021EL | |
| Chemical compound, drug | Methanol | Sigma | Cat# 34860 | |
| Chemical compound, drug | Acetonitrile | Sigma | Cat# 34851 | |

*Continued on next page*

*Continued*

| Reagent type (species) or resource | Designation | Source or reference | Identifiers | Additional information |
|---|---|---|---|---|
| Chemical compound, drug | Human GM-CSF | Celtic Diagnostics | Cat# 300-03-100 | |
| Chemical compound, drug | Histopaque 1077 | Sigma | Cat# 10771 | |
| Chemical compound, drug | DPBS | Lonza | Cat# BE17-512F | |
| Chemical compound, drug | DMSO | Sigma | Cat# 41639 | |
| Chemical compound, drug | Bradford Dye | BIO-RAD | Cat# 500–0205 | |
| Chemical compound, drug | Middlebrook 7H11 Agar | BD | Cat# 283810 | |
| Chemical compound, drug | Polymyxin B | Sigma | Cat# P1004 | |
| Chemical compound, drug | Amphotericin B | Sigma | Cat# A4888 | |
| Chemical compound, drug | Carbenicillin | Sigma | Cat# C1389 | |
| Chemical compound, drug | Trimethoprim | Sigma | Cat# T7883 | |
| Chemical compound, drug | Cell-Tak™ | Corning | Cat# 354241 | |
| Chemical compound, drug | Phorbol 12-myristate 13-acetate | Sigma | Cat# P8139 | |
| Chemical compound, drug | Agarose | Sigma | Cat# A9539 | |
| Chemical compound, drug | SyBr Safe | Invitrogen | Cat# 533102 | |
| Chemical compound, drug | Middlebrook OADC | BD | Cat# 212240 | |
| Chemical compound, drug | RPMI1640 | Lonza | Cat# BE12-167F | |
| Chemical compound, drug | 2-Mercaptoethanol | Gibco | Cat# 21985023 | |
| Chemical compound, drug | 2-Deoxy-D-Glucose | Sigma | Cat# D6134 | |

*Continued on next page*

*Continued*

| Reagent type (species) or resource | Designation | Source or reference | Identifiers | Additional information |
|---|---|---|---|---|
| Chemical compound, drug | D-Glucose $^{13}C6$ | LC Scientific | Cat# GG601L | |
| Chemical compound, drug | Middlebrook 7H9 Broth | BD | Cat# 271310 | |
| Chemical compound, drug | $MgSO_4$ | Sigma | Cat# SAAR41 23920EM | |
| Chemical compound, drug | $NaH_2PO_4$ | Sigma | Cat# S9638 | |
| Software, algorithm | Wave desktop software | Agilent | Version 2.6 | |
| Software, algorithm | GraphPad Prism | Graphpad Prism | Version 7.04 | |
| Software, algorithm | CorelDRAW | Corel | Version X8 | |
| Software, algorithm | XF Cell Mito Stress Test Report Generator | Agilent | | https://www.agilent.com/en/products/cell-analysis/xf-cell-mito-stress-test-report-generator |
| Software, algorithm | XF Glycolysis Stress Test Report Generator | Agilent | | https://www.agilent.com/en/products/cell-analysis/xf-glycolysis-stress-test-report-generator |
| Software, algorithm | XF Glycolytic Rate Assay Report Generator | Agilent | | https://www.agilent.com/en/products/cell-analysis/xf-glycolytic-rate-assay-report-generator |
| Software, algorithm | XF Mito Fuel Flex Test Report Generator | Agilent | | https://www.agilent.com/en/products/cell-analysis/report-generator-for-the-xf-mito-fuel-flex-test |
| Software, algorithm | XF Real-Time ATP Rate Assay Report Generator | Agilent | | https://www.agilent.com/en/products/cell-analysis/xf-real-time-atp-rate-assay-report-generator |

## Ethics Statement

Human monocytes were isolated from buffy coats bought from the South African National Blood Service with approval from SANBS Human Research Ethics Committee (Clearance Certificate No. 2016/02).

## Cell culture

THP-1 cells (ATCC TIB-202) were cultured in RPMI1640 [final concentrations: 4.5 g/L glucose, 2 mM L-GlutaMAX$^{TM}$, 10 mM HEPES and 1 mM sodium pyruvate] containing 10% (v/v) FBS and 0.05 mM 2-mercaptoethanol. THP-1 monocytes were terminally differentiated using 100 nM phorbol 12-myristate 13-acetate (PMA) for 24 h.

Peripheral blood mononuclear cells (PBMC's) were isolated from buffy coats (SANBS) using density gradient centrifugation by overlaying buffy coats, diluted two-fold in phosphate buffered saline (PBS), over Histopaque 1077. CD14$^+$monocytes were isolated using magnetic activated cell sorting (MACS, Militenyi Biotec) and differentiated into macrophages using 10 ng/ml GM-CSF for 6 days in RPMI1640 [10% (v/v) human serum, 1 mM sodium pyruvate (Sigma), 2 mM GlutaMAX$^{TM}$ (Thermo fisher), 1X non-essential amino acids and 10 mM HEPES]. All cell cultures were incubated at 37 °C with 5% $CO_2$. Mycoplasma contamination of the THP-1 cell line was evaluated by PCR using the LookOut Mycoplasma PCR detection kit (Sigma) as per manufacturer's instructions. The identity of the THP-1 cell line was confirmed by Inqaba Biotec (Pretoria, South Africa). Short tandem repeat (STR) sequences were amplified using the SureID21G Human STR identification kit and analysis performed on an ABI PRISM 3500xl genetic analyser. The generated STR profile was validated using the ATCC STR profile for the THP-1 cell line.

## Mycobacterium culture and infection

*Mycobacterium tuberculosis* (H37Rv) obtained from BEI Resources (NR-123), *Mycobacterium bovis* (Danish BCG vaccine strain 1331) from Stratens Serum Institut and *Mycobacterium smegmatis* (mc$^2$155) from Albert Einstein College of Medicine were cultured in Middlebrook 7H9 media containing 0.2% (v/v) glycerol, 10% (v/v) OADC and 0.01% (v/v) tyloxopol. To infect macrophages, the mycobacteria were pelleted at 3 000 x g for 5 min, the supernatant was discarded, and the mycobacteria were resuspended in dPBS and sonicated (3 × 30 s). The optical density at 600 nm (OD$_{600}$) was measured to determine the concentration of mycobacteria and the appropriate volume of mycobacteria required for each MOI was calculated based on the number of monocytes that were seeded into the well. Dead *Mtb* was prepared by heating a known concentration of *Mtb* at 80°C for 20 min.

To determine the percentage of cells infected, an *Mtb* GFP-reporter strain (H37Rv AYs330) was used to infect THP-1 and hMDM cells to determine the percentage of infected cells at increasing MOIs. The *Mtb* GFP-reporter strain contains an episomal construct that constitutively expresses GFP and contains a kanamycin resistance gene (25 µg/ml). The mycobacteria were filtered through a 10 µm syringe filter before infection of macrophages. To assess the presence of extracellular mycobacteria in the washes of the infected macrophages, the final wash was plated out on 7H11 agar plates supplemented with polymyxin B (200 units/ml), amphotericin B (10 µg/ml), carbenicillin (50 µg/ml) and trimethoprim (20 µg/ml) to determine the colony forming units (CFUs) of *Mtb* and BCG.

## Microscopy imaging of macrophages infected with *Mtb*-GFP reporter strain

THP-1 and human monocytes were seeded at 500 000 cells/well into a MatTek 24-well glass-bottom plate and differentiated as described above. The macrophages were infected with *Mtb*-GFP at MOIs of 1, 2.5 and 5 for 16 h. The infected macrophages were washed three times with dPBS to remove most of the extracellular mycobacteria and 300 µl of dPBS was added to the wells. Cells infected with *Mtb*-GFP were viewed on a Nikon Eclipse Ti fluorescent microscope using a FITC filter at 15X magnification. Percentage of infected cells was then calculated by counting the total number of macrophages and the infected macrophages in five fields. A minimum of 100 cells was counted in each field.

## Extracellular flux analysis

Oxygen consumption and extracellular acidification rates (OCR and ECAR, respectively) were measured using the Seahorse XF96 extracellular flux analyzer (Agilent, Santa Clara, CA). Cells were seeded into the XF96 cell culture plate at cell densities of 80000 (hMDM) and 100000 (THP-1) cells per well to generate a confluent monolayer of cells. THP-1 and HMDMs were differentiated directly in the XF96 cell culture plate.

To assess the contribution of the extracellular bacteria to the OCR or ECAR of the infected macrophages, mycobacteria were adhered to the XF cell culture microplate by adding 10 µl of Cell-Tak reagent (135 µl Cell-Tak, 270 µl $dH_2O$, 810 µl of 0.1 N sodium bicarbonate (pH 8.0), followed by adjusting the pH to 7.2–7.8 with 1 N NaOH) to each well. Cell-Tak was incubated for 30 min at 37°C and then the wells were washed three times with dPBS.

Six different assays were performed using the XF96 (*Figure 1*): mitochondrial respiration assay, extracellular acidification assay, glycolytic rate assay, mitochondrial fuel assay, fatty acid oxidation assay and a real-time ATP-Rate production assay (*Figure 1*). The Wave Desktop 2.6 Software, available from the Agilent website (https://www.agilent.com/en/products/cell-analysis/software-download-for-wave-desktop), was used to normalize the OCR and ECAR values to the protein content of each well (see 'Protein Quantification' below) and export the data into GraphPad Prism Version 7.04 for statistical analysis. The Wave Desktop 2.6 software also has the capability to export the data into the XF Report Generators for calculation of the parameters from the specific assays including the mitochondrial respiration assay (Cell Mito Stress Test), extracellular acidification assay (Glycolysis Stress Test), Glycolytic Rate Assay and mitochondrial fuel assay (Mito Fuel Flex Test). The XF Real-time ATP Rate Report generator is available from the Agilent website (https://www.agilent.com/en/products/cell-analysis/xf-real-time-atp-rate-assay-report-generator).

In the mitochondrial respiration assay (*Figure 1B*), three or four OCR measurements (3 min of mixing, 4 min of measurement) were performed under basal conditions to determine basal respiration, followed by the addition of oligomycin, an inhibitor of ATP synthase, and another three OCR measurements to determine the ATP-linked OCR and proton leak. FCCP, an ETC uncoupling agent, was injected to determine the maximal respiration and the spare respiratory capacity, which is the difference between the maximal and basal respiration. Lastly, rotenone (inhibitor of complex I) and antimycin A (inhibitor of complex III) were injected to inhibit the electron transport chain to determine non-mitochondrial respiration, which is subtracted from the basal measurements to give basal respiration, the ATP-linked OCR to give the proton leak and the maximal OCR measurements to give maximal respiration (*Figure 1B*).

Final concentrations of mitochondrial modulators in the XF96 well were 1 µM for carbonilcyanide *p*-triflouromethoxyphenylhydrazone (FCCP) for both THP-1 and hMDMs; 1.5 µM for oligomycin for all cell types; and 0.5 µM each of antimycin A and rotenone used in combination for THP-1 cells and 2.5 µM each for hMDMs. Other reagents were used as per the manufacturer's instructions (Agilent, Santa Clara, CA).

After the respective differentiation and treatment/infection periods, growth medium was removed, and the cells were washed twice with XF assay medium prior to adding 180 µl of XF assay medium for the XF run. In the case of the mitochondrial respiration test and the extracellular acidification test, the assay medium used was DMEM supplemented with 1 mM sodium pyruvate and 2 mM GlutaMAX. The media for the mitochondrial respiration test was supplemented with 25 mM glucose.

In the extracellular acidification test (*Figure 1C*), the first three basal ECAR readings (2 min of mixing and 3 min of measurement) were used to measure the non-glycolytic acidification. After addition of 10 mM glucose, three ECAR measurements were used to determine the extracellular acidification resulting from glucose metabolism, after subtraction of the non-glycolytic acidification. This was followed by addition of oligomycin to inhibit the ATP synthase in the mitochondria, forcing the cell to use glycolysis solely to generate all the ATP demands of the cell. Three subsequent ECAR measurements gave a measure of the maximal glycolytic capacity after subtraction of the non-glycolytic acidification. The difference between the glycolytic capacity and the ECAR from glucose metabolism gives the glycolytic reserve, which is a measure of the capability of a cell to respond to energetic demands in addition to indicating how close the ECAR from glucose metabolism is to the cell's theoretical maximum. Glycolysis was then inhibited with addition of 2.5 mM 2-deoxyglucose (2-DG) and three subsequent ECAR measurements demonstrated that the prior ECAR resulted from glycolysis, hence giving a measure for non-glycolytic acidification.

For the fatty acid oxidation assay (*Figure 1F*), growth media were exchanged for substrate-limited media (DMEM supplemented with 0.5 mM glucose, 1 mM GlutaMax, 0.5 mM carnitine (Sigma) and 1% FBS) 16 h prior to the assay. Forty-five minutes prior to the assay, substrate limited media were exchanged for the fatty acid oxidation assay medium (111 mM NaCl, 4.7 mM KCl, 1.25 mM $CaCl_2$, 2 mM $MgSO_4$, 1.2 mM $NaH_2PO_4$ supplemented with 2.5 mM glucose, 0.5 mM carnitine and 5

mM HEPES on the day of the assay). Fifteen min prior to the assay, etomoxir (Sigma) was added to the respective wells to a final concentration of 40 µM. XF palmitate-bovine serum albumin (BSA) conjugate or BSA control was added to the appropriate wells prior to loading the XF cell culture plate into the XF to perform a mitochondrial respiration assay.

For the mitochondrial fuel test (*Figure 1E*), growth media was exchanged with XF base medium (Agilent) supplemented with 1 mM sodium pyruvate, 2 mM GlutaMax and 10 mM glucose. Final concentrations of inhibitors used in the mitochondrial fuel test were: 3 µM BPTES (Gln oxidation inhibitor), 16 µM etomoxir (long-chain fatty acid oxidation inhibitor) and 2 µM UK5099 (glucose oxidation inhibitor). The drug combinations were prepared and loaded into the ports of the XF cartridge as described for the Flexibility Test in the *Agilent Seahorse XF Mito Fuel Flex Test Kit User Manual.* Each infection was compared with uninfected macrophages as the control in separate XF runs.

For the glycolytic rate assay and the real-time ATP-rate production assay, growth media were exchanged with XF base medium, without phenol red, supplemented with 1 mM sodium pyruvate, 2 mM GlutaMAX$^{TM}$, 10 mM glucose and 5 mM HEPES.

In the glycolytic rate assay (*Figure 1D*), three initial OCR and ECAR measurements (3 min of mixing, 4 min of measurement) were used to determine the total proton efflux rate (PER). This was followed by simultaneous addition of antimycin A and rotenone to inhibit the electron transport chain (OXPHOS) and consequently the TCA cycle that results in the carbonic acid produced from the $CO_2$ generated by the TCA cycle. The next three OCR and ECAR readings were measured to determine the mitoPER (PER from mitochondrial respiration) and compensatory glycoPER (PER from glycolysis). Finally, 2-DG was injected to inhibit glycolysis and provide qualitative confirmation that the PER prior to the 2-DG injection results from glycolysis.

In the real-time ATP rate assay (*Figure 1G*), after three OCR and ECAR readings to determine total PER, oligomycin was added to inhibit ATP synthase in the mitochondria. The next three OCR readings were used to determine the rate of oxygen consumed for ATP production in the mitochondria. Antimycin A and rotenone were then injected simultaneously to inhibit the extracellular acidification from mitochondrial respiration to determine glycoPER which is equivalent to the glycolytic ATP rate of production.

All drugs for the respective assays were prepared at 10X their desired concentration and loaded into the XF cartridge into the respective ports. All assay media were pre-warmed to 37 ˚C and pH corrected to 7.4 (mitochondrial respiration test, mitochondrial fuel test, and fatty acid oxidation assay, glycolytic rate assay) and 7.35 (extracellular acidification assay).

## Metabolite extraction

THP-1 ($3 \times 10^6$ cells per well) or hMDMs ($3 \times 10^6$ cells per well) were seeded into six well plates and differentiated as previously described. Cells were infected with the appropriate mycobacterium strain for a total of 18 h. For $^{13}C_6$-glucose labelling of metabolites, media were prepared by adding 4.5 g/l of $^{13}C_6$-glucose to hMDM or THP-1 media containing glucose-free RPMI 1640 (Glucose free, Sigma). Initially, uninfected or infected cells were incubated in normal unlabelled media for 10 h. Then the cells were washed three times with glucose-free media before adding RPMI 1640 containing $^{13}C_6$-glucose. Cells were incubated for a further 8 h to allow incorporation of the radiolabelled substrate. Then the supernatants of the cells were collected and the cells were washed in pre-warmed (37 ˚C) PBS three times prior to adding ice-cold methanol to quench the metabolism of the cells. An equal amount of ice-cold water containing internal standards of 6-amino-nicotinic acid and nitrophenyl phosphate (to give a final concentration of 700 ng/ml) was added, and cells were lifted into the methanol-water using a cell scraper. The cell suspensions were transferred to 2 ml screw-cap tubes and metabolites were released by three freeze-thaw cycles followed by a 3000 x g centrifugation for 3 min to remove the cell debris. Metabolites were transferred to Spin-X centrifuge tube filters (Sigma) and centrifuged at 3000 x g for 5 min. To remove the methanol, the filtrate was evaporated at 45 ˚C for approximately 16 h and the lyophilized metabolites were resuspended in 100 µl distilled water and transferred to mass spectroscopy snap ring vials (Separations) for metabolite analysis.

The collected supernatants were transferred to Spin-X centrifuge tube filters (Sigma) and centrifuged at 3000 x g for 5 min. The filtrate was evaporated at 45 ˚C for approximately 16 h and the lyophilized metabolites were resuspended in 100 µl distilled water for LC-MS/MS analysis.

For amino acid analysis, 50 µl of the metabolite extraction was diluted with 50 µl of acetonitrile.

## LC-MS/MS

Both internal metabolites as well as excreted metabolites in the supernatant fluid (labelled or unlabelled) were analysed using LC-MS/MS.

Metabolite quantification was measured using an AB Sciex 5500 Q-Trap, triple quadrupole mass spectrometer in multiple reaction monitoring mode. A Waters XTerra C18 column (2.1 mm X 50 mm, 3.5 µM) was used at 40 ˚C with a mobile phase of A: 10 mM tributylamine, 5 mM acetic acid, 1% acetonitrile in double-distilled water and mobile phase B of 10 mM tributylamine, 5 mM acetic acid in acetonitrile. A gradient separation was employed: T0 min - 0%B, T1 min - 0%B, T10 min - 100%B, T15 min-100%B, T15.1 min - 0%B, T25 min - 0%B.

$^{13}C_6$-glucose incorporation was done using a Thermo Q Exactive working in full MS negative ion mode with a scan range of 50–750 m/z, resolution 70000 and AGC target 1e6. A Bio-Rad Aminex HPX-87H, 300 mm X 7.8 mm column was used with an isocratic mobile phase of Milli-Q water with 0.1% formic acid.

Amino acid analysis was performed on a Q Exactive MS coupled to a Dionex 3000 UPLC system, using a Waters Acquity BEH amide column, 2.1 mm x 100 mm, with a 1.7 µM particle size. The MS was operated in positive MS full-scan mode with a resolution of 70000 from 50 to 750 m/z and an AGC target of 1e6. Separation was achieved using mobile phase A: Double-distilled water with 0.1% formic acid and B: LCMS grade acetonitrile with 0.1% formic acid. A HILIC mode gradient with the following characteristics was used: T0 min – 99%B, T0.2 min – 99%B, T24 min – 30%B, T25 min – 30%B, T25.1 min – 99%B, T35 min – 99%B.

## Protein quantification

OCR, ECAR and metabolite levels were normalized to the amount of protein present in the well. This was done directly in the XF cell culture plate (OCR/ECAR) or in a microfuge tube (metabolites) using a Bradford dye (Bio-Rad) as per the manufacturer's instructions. Briefly, for normalizing OCR and ECAR data, the supernatant was removed leaving behind 10 µl to prevent aspiration of cells and an equal amount of formalin was added to kill remaining Mtb to remove samples from the BSL3. These cells were lysed using NaOH at a final concentration of 12.5 mM. Protein standards (Bovine Serum Albumin: 0.125, 0.25, 0.5, 0.75, 1, 1.5 and 2 mg/ml) were added to the respective wells, treated with formalin and 12.5 mM NaOH similarly to the samples and Bradford dye was added (150 µl) to all the wells. The plate was incubated for 5 min in the dark and absorbance at 595 nm measured using a Synergy H4 Hybrid reader (Biotek).

For normalization of metabolite data, cells were scraped and lysed (freeze-thawed) in formalin. This was evaporated (45 ˚C, overnight) and resuspended in distilled water. Bradford reagent was added to these samples, transferred to a 96 well microplate and incubated for 5 min in the dark prior to reading at 595 nm.

## Statistical analysis

All experiments were performed independently twice or more with a minimum of six biological replicates each. The data are presented as the mean and SD of one representative independent experiment (unless otherwise stated in the figure legends). Analysis was performed in Graphpad Prism Version 7.04. One-way Anova was used for multiple comparisons and Students t-test was used to compare pairs. Further details are given in the figure legends.

## Acknowledgements

This work was supported by NIH grants R01AI111940, R21127182 and R61AI138280, DOD Discovery Award PR121320, a Bill and Melinda Gates Foundation Award (OPP1130017) and pilot funds from the UAB Centers for AIDS Research and Free Radical Biology and UAB School of Medicine Infectious Diseases and Global Health and Vaccines Initiative to AJCS. The research from which this publication emanated was also co-funded by the South African Medical Research Council. AJCS is a Burroughs Welcome Investigator in the Pathogenesis of Infectious Disease. We also acknowledge Katya Govender for the LC-MS/MS analysis of our samples, and Duran Ramsuran and Vanisha Munsamy for assisting with the fluorescent imaging to determine the percentage infected cells.

## Additional information

### Funding

| Funder | Grant reference number | Author |
|---|---|---|
| National Institutes of Health | R01AI111940 | Adrie JC Steyn |
| U.S. Department of Defense | PR121320 | Adrie JC Steyn |
| National Institutes of Health | R21127182 | Adrie JC Steyn |
| South African Medical Research Council | | Adrie JC Steyn |
| Bill and Melinda Gates Foundation | OPP1130017 | Adrie JC Steyn |
| National Institutes of Health | R61AI138280 | Adrie JC Steyn |
| UAB Centres for AIDS Research and Free Radical Biology | Pilot Funds | Adrie JC Steyn |
| UAB School of Medicine Infectious Diseases, Global Health and Vaccines Initiative | Pilot Funds | Adrie JC Steyn |

The funders had no role in study design, data collection and interpretation, or the decision to submit the work for publication.

### Author contributions

Bridgette M Cumming, Conceptualization, Formal analysis, Supervision, Investigation, Methodology, Writing—original draft, Writing—review and editing; Kelvin W Addicott, John H Adamson, Formal analysis, Methodology; Adrie JC Steyn, Conceptualization, Supervision, Funding acquisition, Writing—review and editing

### Author ORCIDs

Bridgette M Cumming (iD) http://orcid.org/0000-0001-6977-765X
Adrie JC Steyn (iD) https://orcid.org/0000-0001-9177-8827

### Ethics

Human subjects: Human monocytes were isolated from buffy coats bought from the South African National Blood Service with approval from SANBS Human Research Ethics Committee (Clearance Certificate No. 2016/02)

### Decision letter and Author response

Decision letter https://doi.org/10.7554/eLife.39169.017
Author response https://doi.org/10.7554/eLife.39169.018

## Additional files

### Supplementary files

• Transparent reporting form
DOI: https://doi.org/10.7554/eLife.39169.015

### Data availability

All data generated or analysed during this study are included in the manuscript and supporting files.

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
