## [Decision Letter]

Thank you for submitting your article "*Mycobacterium tuberculosis* induces decelerated bioenergetic metabolism in human macrophages" for consideration by *eLife*. Your article has been reviewed by Gisela Storz as the Senior Editor, a Reviewing Editor, and three reviewers. The following individuals involved in review of your submission have agreed to reveal their identity: Dirk Bald (Reviewer #2); William R Jacobs (Reviewer #3).

The reviewers have discussed the reviews with one another and the Reviewing Editor has drafted this decision to help you prepare a revised submission.

Summary:

This manuscript describes a bioenergetics analysis of macrophages infected with *Mycobacterium tuberculosis* using extracellular flux technology. The authors test various parameters related to energy metabolism and carbon utilization in different macrophages infected with *M. tuberculosis*. The intent is to describe potential avenues in which *Mycobacterium tuberculosis (Mtb*) rewires the metabolism profile of various macrophage lines to create a favorable niche allowing *Mtb* to thrive in a normally unfavorable environment.

Key findings:

1) *M. tuberculosis* infection results in reduced glycolysis and decreased flux through the citric acid cycle, an effect that is not seen with BCG or dead mycobacteria.

2) Infection with tubercle bacteria leads to increased fatty acid utilization by mitochondria and a shift towards a quiescent phenotype. ATP production is also reduced in mitochondria upon *M. tuberculosis* infection.

3) The source of fatty acids used also shifts from endogenous to exogenous upon *M. tuberculosis* infection.

Essential revisions:

1) The study needs to be framed in the context of existing literature, for example the study by Lachmandas et al. (2016). Whilst this current submission seems to be the first to apply extracellular flux analysis using a high-throughput automated system, others have reported on rewiring of energy (and primary carbon) metabolism in the host upon *M. tuberculosis* infection. Consider this as you frame the background to your work. Providing the correct context, with a careful exposition of all the relevant material allows readers to more easily digest your work.

2) All reviewers concurred that the overall readability and presentation your manuscript is poor. The use of jargon obfuscates the findings and reduces impact. Methods are not described with sufficient detail and provide confusing information for example, different durations of infections 18 versus 10 hours (subsection “Metabolite extraction”). Biological replicates, data analysis etc. are not clear. Note that *eLife* has a broad readership and the expectation is that manuscripts reach as many people as possible. Please consider this and recraft your presentation of the work to provide a clearer description. Good use of figures and diagrams can be very instructive.

3) Experimental weaknesses include:

i) What proportion of cells remain uninfected? This needs to be considered as these also contribute to the readout. Please consider a way of addressing this, it is an important point.

ii) Were there any dead host cells (and did the proportion of these change with time)? Again, very important as you would be looking at the average readout of the whole cellular population in the well during any given experiment.

iii) Did you confirm that there were no extracellular organisms? Regards point (ii), will it be useful if you describe flux analysis with a lower MOI than that reported? Using the same MOI throughout the paper will give your claims more support.

4) All reviewers raised concerns with the glycolysis effect and indicated that it needs to be characterized more carefully. Decreased glycolytic flux is supported by data shown in the histogram in Figure 3D. However, Figure 3C shows increased ECAR and a stronger ECAR increase after glucose addition for the *Mtb*-infected macrophages compared to uninfected control. This point needs to be clarified. Also, it would be important to confirm that the effects seen are due solely to the mycobacterial infected cells and not due to the addition of the glucose (also referring to the point above regards uninfected cells). Comparing AKT production/phosphorylation is important.

5) More controls (that provide no measurable difference in the readout) are needed. Things like differences in the basal respiration rate need to be clarified. The effect of adding dead mycobacteria is unclear, why was this experiment necessary? Clarify.

---

## [Author Response]

Essential revisions:1) The study needs to be framed in the context of existing literature, for example the study by Lachmandas et al. (2016). Whilst this current submission seems to be the first to apply extracellular flux analysis using a high-throughput automated system, others have reported on rewiring of energy (and primary carbon) metabolism in the host upon M. tuberculosis infection. Consider this as you frame the background to your work. Providing the correct context, with a careful exposition of all the relevant material allows readers to more easily digest your work.

We thank the reviewers for this helpful comment. As correctly pointed out, others have indeed reported on rewiring of metabolism in the host upon *Mtb* infection. However, a major shortfall of the study by Lachmandas et al. is that all the experiments on PBMC or monocytes were performed using stimulation with an *Mtb* H37Rv lysate and not live virulent *Mtb* as in our study. In the Lachmandas et al. article, please refer to the paragraph entitled “Stimulation” under “Materials and Methods” on page 2581 (Lachmandas et al., 2016). In addition, another study investigating the rewiring of metabolism in *Mtb* infected macrophages, Gleeson et al. (2016) irradiated (killed) *Mtb* H37Rv (indicated by “iH37Rv” in Figure 1) or the attenuated strain H37Ra was used as indicated in the figure legends of Figure 2, Figure 3 and Figure 4 (Gleeson et al., 2016). The reason for infection using *Mtb* lysates, irradiated H37Rv, or attenuated H37Rv is that these strains/preparations are non-pathogenic and therefore the experiments can be conducted outside a BSL3 environment. For this reason, we have included dead (heat killed) *Mtb* H37Rv as control in our infection studies to demonstrate how the metabolic rewiring induced by live *Mtb* H37Rv differs to that of dead *Mtb* and to underscore the need to interpret findings from experiments using dead *Mtb* or lysate infected macrophages with caution.

Nonetheless, we appreciate the reviewers comment and to provide context to our study, we have now included a paragraph of existing literature (Introduction).

2) All reviewers concurred that the overall readability and presentation your manuscript is poor. The use of jargon obfuscates the findings and reduces impact. Methods are not described with sufficient detail and provide confusing information for example, different durations of infections 18 versus 10 hours (subsection “Metabolite extraction”). Biological replicates, data analysis etc. are not clear. Note that eLife has a broad readership and the expectation is that manuscripts reach as many people as possible. Please consider this and recraft your presentation of the work to provide a clearer description. Good use of figures and diagrams can be very instructive.

“The use of jargon obfuscates the findings and reduces impact.”: We apologize for the reviewers comments. However, we are having difficulty assessing which words are “jargon” as we are using terminology that is customary to the bioenergetic field. We have carefully scrutinized our manuscript and identified words that could be perceived as “jargon” including, but not limited to: Spare respiratory capacity (SRC), ATP-linked OCR, proton leak, non-mitochondrial respiration, glycolytic proton efflux rate (PER; glycoPER), ATP production rate, compensatory glycolytic proton efflux rate, maximum respiration (Max Resp), glycolytic reserve, metabolic plasticity, metabolic health, non-glycolytic extracellular acidification, bulk acidification, bioenergeic phenogram, energetic phenotype, quiescent phenotype, mitochondrial flexibility and dependancy. However, these words are well-established terms in the bioenergetic field (Abildgaard et al., 2014; Keuper et al., 2014; Lim et al., 2015; Pisani et al., 2018; Ryan et al., 2018; van der Windt et al., 2012) of which many are used in the Agilent Seahorse XF assay User Manuals: (https://www.agilent.com/cs/library/usermanuals/public/103344-400.pdf; https://www.agilent.com/cs/library/usermanuals/public/103592-400_Seahorse_XF_ATP_Rate_kit_User_Guide.pdf

https://www.agilent.com/cs/library/usermanuals/public/XF_Glycolysis_Stress_Test_Kit_User_Guide.pdf

https://www.agilent.com/cs/library/usermanuals/public/XF_Cell_Mito_Stress_Test_Kit_User_Guide.pdf

https://www.agilent.com/cs/library/usermanuals/public/XF_Palmitate_BSA_Substrate_Quickstart_Guide.pdf

https://www.agilent.com/cs/library/usermanuals/public/XF_Mito_Fuel_Flex_Test_Kit_User_Guide%20old.pdf

Regardless, we followed the reviewers’ advice and have now explained the assays and their parameters in much more detail in the subsection “Extracellular Flux Analysis”. Furthermore we significantly improved Figure 1, which now includes equations defining how the parameters were calculated to improve readability.

“Methods are not described with sufficient detail and provide confusing information for example, different durations of infections 18 versus 10 hours (subsection “Metabolite extraction”).”: Regarding the carbon-tracing experiments referred to by the reviewers in subsection “Metabolite extraction”, we have now expanded the description to make this clearer. Briefly, the total infection time for the carbon-tracing experiments was 18 hours. However, we only allowed incorporation of the radiolabeled glucose in the last 8 hours of infection to limit the number of cycles through the TCA cycle into which the radiolabeled carbon was incorporated to facilitate carbon-tracing.

“Biological replicates, data analysis etc. are not clear.”: We thank the reviewers for the comments concerning the biological replicates and data analysis. However, in the original manuscript, we have indicated the number of independent experiments performed and the statistical analysis used in each figure legend to generate the p-values that were depicted in the Figure. Furthermore, in the original Materials and methods section, we have a paragraph entitled “Statistical analysis”, where we indicated that each independent experiment was performed with a minimum of six biological replicates. This refers to all the assays and experiments conducted for the manuscript (subsection “Protein Quantification**”**). We have now added the description “biological replicates” to our n values in the legend of each figure.

3) Experimental weaknesses include:i) What proportion of cells remain uninfected? This needs to be considered as these also contribute to the readout. Please consider a way of addressing this, it is an important point.ii) Were there any dead host cells (and did the proportion of these change with time)? Again, very important as you would be looking at the average readout of the whole cellular population in the well during any given experiment.iii) Did you confirm that there were no extracellular organisms? Regards point (ii), will it be useful if you describe flux analysis with a lower MOI than that reported? Using the same MOI throughout the paper will give your claims more support.

We thank the reviewers for these comments and have addressed the concerns below.

“i) What proportion of cells remain uninfected?”:The proportion of uninfected cells were determined by infecting macrophages with *Mtb*-GFP at the same MOIs and experimental conditions as used for the XF assays. Using bright field and fluorescence microscopy, we counted the uninfected and infected cells and determined the percentage of cells that were infected. The percentage of infected cells increased as the MOI was increased (Figure 2—figure supplement 2E-G), from ± 50% at an MOI of 1 to more than 80% at an MOI 5. Although the percentage of uninfected cells will contribute to the readout of the XF profiles, previous studies have demonstrated that lipids shed by intracellular mycobacteria, such as TDM and PIM2, spread via the endocytic network throughout the macrophage, and via exocytic vesicles to neighboring uninfected cells (Beatty et al., 2000; Xu et al., 1994) and can elicit the production of proinflammatory cytokines by these neighbouring cells (Rhoades et al., 2003). Consequently, the bioenergetic metabolism of the “by-stander” uninfected cells will also be modulated. Thus, the profiles of increasing MOI are providing an overview of the bioenergetics at different stages of infection. We have explained this in the Results section.

“ii) Were there any dead host cells (and did the proportion of these change with time)?”:The reviewers are correct. Some infected macrophages do die over the time of infection; however, as we allow the monocytes to naturally adhere to the base of the wells of the XF96 cell culture microplate prior to differentiation and infection, the dead macrophages will naturally lift from the surface of the wells. Prior to the XF run, the adhered cells are washed twice with XF media, thus removing the lifted dead cells prior to the XF run. Hence, the dead cells will not contribute to the average readout of the whole cellular population that is adhered to the base of the well. Furthermore, to mitigate the differences in the cell number that are adhered to the base of each well and the effects of these differing numbers on the basal respiration (OCR) or ECAR, we normalized the OCR and ECAR measurements of each well to the protein content of each well, which was determined after the XF run. This is reflected in the units of the Y-axis of the XF profiles.

“Regards point (ii)[…]”: When we investigated MOIs lower than 1, negligible differences were observed in the XF profiles between the uninfected and infected cells after 24 hours of infection. Longer periods of infection in the XF cell culture microplates are problematic as culturing the high density of infected macrophages in the confluent monolayer required for XF analysis would require changing the media every 3-4 hours. This will require removing the cells from the optimal 37°C, 5% CO_2_ environment of the incubator on a regular basis to change the media and changing the media could increase the risk of lifting the *Mtb* infected cells. We seed the monocytes and differentiate them directly in the wells of the XF96 microtiter plate as we have previously experienced excessive loss of viable macrophages when they were differentiated and infected in a separate flask and then lifted to reanneal to the XF96 cell culture microplate. For these reasons, we have used a MOI of 1 as the lowest MOI investigated.

With reference to using the same MOI throughout the paper, we have used a MOI of 5 in all the XF profiles illustrated in the paper, with the exception of the results of the real-time ATP rate assay (Figure 5F and 5G). The XF profiles of lower MOIs of 1 and 2.5 are illustrated in the Supplementary information. The reason for using several MOIs in our experiments is that it substantially improves the critical assessment and interpretation of our data, which otherwise, would have been severely limited. Also, since multiple MOI’s represent multiple bacillary burden, we anticipated this to be an obvious question to many investigators in the microbial pathogenesis field. Lastly, we also wanted to demonstrate the sensitivity of the XF to detect differences observed in vitro between initial and more advanced stages of infection. Overall, we believe it represents good science.

“iii) Did you confirm that there were no extracellular organisms?”: Yes, we did indeed examined the presence of extracellular organisms. To remove any extracellular bacteria prior to the XF run, we removed the culture medium and washed the adhered infected macrophages twiRhoades et al., 2003ce with the XF media prior to adding the XF media in preparation for the XF run. This was also standard protocol with the XF run to ensure that all the plasma and culture media was removed from the well prior to the XF run. In response to the reviewers’ comment, we plated out the XF media after the two washes of the macrophages on 7H11 agar plates. Less than 200 CFU were obtained per well from the washes of the infected hMDMs (MOI 5), and less than 100 CFU per well from the washes of the infected THP-1 cells (MOI 5). Lower CFU counts were obtained at the lower MOIs. Also, to demonstrate that these extracellular mycobacteria do not contribute to the host OCR or ECAR, the washes were transferred to a separate XF cell culture microplate and a separate mitochondrial respiration assay was performed on any extracellular bacteria present in the washes. The OCR readings obtained were below 0 pmol/min and the ECAR readings were equivalent to 0 mpH/mol, and the extracellular bacteria did not respond to the sequential injections of oligomycin, FCCP, rotenone and antimycin A. Thus, the few extracellular bacteria remaining in the well do not contribute to the OCR of the macrophages during the XF run. We have now included these data in the revised manuscript (Figure 2—figure supplement 2A-D).

4) All reviewers raised concerns with the glycolysis effect and indicated that it needs to be characterized more carefully. Decreased glycolytic flux is supported by data shown in the histogram in Figure 3D. However, Figure 3C shows increased ECAR and a stronger ECAR increase after glucose addition for the Mtb-infected macrophages compared to uninfected control. This point needs to be clarified. Also, it would be important to confirm that the effects seen are due solely to the mycobacterial infected cells and not due to the addition of the glucose (also referring to the point above regards uninfected cells). Comparing AKT production/phosphorylation is important.

We thank the reviewers for bringing this discrepancy to our attention. Regrettably, we inadvertently inserted the incorrect ECAR XF profile in Figure 3C, hence, we appreciate the reviewers meticulous attention to detail. We have now included the correct XF profile in Figure 3C, which does correlate with the findings discussed in the manuscript as depicted in the histogram in Figure 3D.

As our manuscript is a resource article demonstrating the use of extracellular flux analysis to examine the bioenergetic metabolism of the *Mtb* infection of the macrophages, we have not investigated mechanisms inducing the observed modulations of the bioenergetic metabolism. For this reason, we respectfully disagree with the reviewer that it would be relevant to our manuscript to compare AKT production/phosphorylation. Lachmandas et al. examined which cells, receptors and regulators promoted the *Mtb*-induced switch to glycolysis in humans as the primary aim of their paper. Their investigations with dead *Mtb* lysate stimulation of peripheral blood mononuclear cells (PBMC) found that the switch in host metabolism toward aerobic glycolysis in human PBMC was mediated *in part* through the activation of AKT-mTOR pathway.

5) More controls (that provide no measurable difference in the readout) are needed. Things like differences in the basal respiration rate need to be clarified. The effect of adding dead mycobacteria is unclear, why was this experiment necessary? Clarify.

We thank the reviewers for these comments, however we do not fully understand this concern. We assume the reviewers are referring to the small differences in the observed basal respiration between experiments. These small differences in basal respiration between experiments arise from the fact that we differentiate the monocytes into macrophages in each XF well rather than differentiating them in bulk, lifting the cells and seeding them into the XF wells. We did investigate this initially, but lifting the infected differentiated macrophages severely affected the viability and bioenergetics of the macrophages. For this reason, the differentiated uninfected macrophages are *always* run as a control in each XF run. Lastly, the XF96 is an extremely sensitive instrument that measures OCR in the range of picomoles (10^-12^) oxygen consumed per minute and ECAR in mpH change per minute, thus any small variations in the OCR or ECAR will be detected at a picomole and mpH level.

There is the further variable of the different donors of the PBMCs from which we isolate the monocytes to differentiate into hMDMs that will also contribute to small differences in basal respiration. As the numbers of monocytes required for the XF assays are very high (80 000 per well per XF96 cell culture microplate), it is not possible to use the same donor for all the experiments. To eliminate donor bias, we combine the monocytes from three different donors for each XF experiment. Donor variation is inevitable and beyond our control.

Several lines of evidence point to dead *Mtb* as an essential control and emphasizes the scientific rigour of our experimentation. Firstly, dead *Mtb* is not virulent as it is unable to produce and secrete virulence factors that cause disease, whereas live *Mtb* is virulent. Secondly, previous studies have used *Mtb* lysate (Lachmandas et al., 2016) or γ-irradiated (dead) *Mtb* (Gleeson et al., 2016), to infect the macrophages, and obtained significantly different findings from our study. Hence, we included the dead *Mtb* control to illustrate the clear differences in the bioenergetic metabolism of macrophages infected with live *Mtb* or dead *Mtb*. Thus, using *Mtb* lysate or γ-irradiated *Mtb* infected macrophages do not give an accurate representation of the metabolic modulations induced by infection with live *Mtb*. Therefore, caution needs to be exercised when interpreting results generated with infection with *Mtb* lysate and γ-irradiated *Mtb*. As requested, we have elaborated on this further in the Introduction and Discussion section. Thirdly, we used the dead *Mtb* control to demonstrate that pharmacological killing of *Mtb* in the macrophages will not necessarily restore the bioenergetic metabolism of the macrophage back to that of the uninfected macrophage. Again, the logical conclusion is that remnants of dead *Mtb* cells are still capable of inducing a bioenergetic response in the macrophage, but distinctly from live *Mtb*.